# Update on the Pathology of Pediatric Liver Tumors: A Pictorial Review

**DOI:** 10.3390/diagnostics13233524

**Published:** 2023-11-24

**Authors:** Priyanka Bhagat, Mukul Vij, Lexmi Priya Raju, Gowripriya Gowrishankar, Jagadeesh Menon, Naresh Shanmugam, Ilankumaran Kaliamoorthy, Ashwin Rammohan, Mohamed Rela

**Affiliations:** 1Department of Pathology, Choithram Hospital and Research Center, Manik Bagh Road, Indore 452014, Madhya Pradesh, India; drpriyankabhagat@gmail.com; 2Department of Pathology, Dr. Rela Institute & Medical Centre, No. 7 CLC Works Road Chromepet, Chennai 600044, Tamil Nadu, India; dr.priyaraju1794@gmail.com (L.P.R.); gowripriya.g@relainstitute.com (G.G.); 3The Institute of Liver Disease & Transplantation, Dr. Rela Institute & Medical Centre, No. 7 CLC Works Road Chromepet, Chennai 600044, Tamil Nadu, India; jagadeeshmenonv@gmail.com (J.M.); drnareshps@gmail.com (N.S.); ilan.kaliamoorthy@relainstitute.com (I.K.); ashwinrammohan@gmail.com (A.R.); mohamedrela@gmail.com (M.R.)

**Keywords:** pediatric liver tumors, pathology, hemangioma, mesenchymal hamartoma, undifferentiated embryonal sarcoma, inflammatory myofibroblastic tumor, hepatoblastoma, hepatocellular carcinoma

## Abstract

Liver tumors in children are uncommon and show remarkable morphologic heterogeneity. Pediatric tumors may arise from either the epithelial or mesenchymal component of the liver and rarely may also show both lines of differentiation. Both benign and malignant liver tumors have been reported in children. The most common pediatric liver tumors by age are benign hepatic infantile hemangiomas in neonates and infants, malignant hepatoblastoma in infants and toddlers, and malignant hepatocellular carcinoma in teenagers. Here, we provide an up-to-date review of pediatric liver tumors. We discuss the clinical presentation, imaging findings, pathology, and relevant molecular features that can help in the correct identification of these tumors, which is important in managing these children.

## 1. Introduction

Liver tumors constitute the third most common solid abdominal tumors in children, after neuroblastoma and Wilm’s tumor [1]. Among the space occupying lesions of the liver, malignant tumors constitute two-thirds of the chunk, with the remaining one-third being hemangiomas, benign cysts, or benign tumors [2]. Liver masses may be detected by the parent while bathing the child, palpated during a clinical examination, or detected upon imaging. The benign liver lesions may be inborn and may increase in size as the child grows. In early childhood, hemangiomas and in adolescents hepatocellular adenomas (HCAs) constitute the majority of benign liver lesions [2]. The other benign lesions encountered in children are mesenchymal hamartoma (MH) and inflammatory myofibroblastic tumors (IMFTs). Among the malignant tumors, hepatoblastoma (HB) and hepatocellular carcinoma (HCC) are the most common tumors in early childhood and adolescents respectively [3]. Other rare malignant tumors include epithelioid hemangioendothelioma, malignant rhabdoid tumor (MRT), hepatobiliary rhabdomyosarcoma (RMS), embryonal sarcoma (ES), and angiosarcoma. HB with carcinoma features (HBC) is a recently introduced malignant tumor. The classification of pediatric liver tumors is shown in Table 1.

## 2. Mesenchymal Tumors in Children

### Hemangioma

Hemangiomas in children may be congenital or infantile, with the infantile type being more common. Figure 1 classifies the types of hepatic hemangioma in children.

## 3. Hepatic Congenital Hemangioma

### 3.1. Clinical Presentation

Hepatic congenital hemangioma (HCH) is a benign vascular tumor that develops in utero and is fully grown at birth, with no post-natal increase in size [4,5,6]. It may rapidly involute at infancy (rapidly involuting congenital hemangioma, RICH), may partially involute (partially involuting congenital hemangioma, PICH), or may remain as it is (non-involuting congenital hemangioma, NICH) [6]. These are primarily asymptomatic tumors found incidentally upon imaging in infancy or upon prenatal ultrasound. Some newborns may present with liver enlargement, low platelets, hypofibrinogenemia, and rarely with high-output cardiac failure from arteriovenous or portohepatic venous shunting [4,7]. Hydrops fetalis has also been reported.

### 3.2. Imaging Findings

HCH occurs most commonly as a solitary lesion in the right hepatic lobe and upon ultrasound showing a large mass with an extensive central infarction, cystic change, hemorrhage, calcifications, and sometimes an arterial feeding vessel that has direct shunts to hepatic veins. The use of magnetic resonance imaging (MRI) and computed tomography (CT) may show lesions with centripetal enhancements [4,7,8]. A diagnosis can be made upon prenatal imaging.

### 3.3. Pathology

Histologically, HCH is composed of vague aggregates or lobules of variously sized capillary vessels lined by a single layer of round to oval benign endothelial cells with inconspicuous nucleoli and few pericytes (Figure 2) [8]. These aggregates are often located at the tumor’s periphery, and the individual capillaries are separated by fibrotic stroma. In RICH, the vessels are thin-walled and the central area where the involution starts may show necrosis, dystrophic calcification, thrombosis, and hemorrhage [5]. Hemosiderin pigment deposition is also demonstrated. Foci of extramedullary hematopoiesis are also noted frequently. NICH, in contrast, lacks involutional changes such as necrosis, hemorrhage, fibrosis, and hemosiderin. PICH shows overlapping features in between RICH and NICH. Upon immunohistochemistry (IHC), the endothelial cells show positivity for CD 31, CD 34, WT 1, factor VIII, and erythroblast transformation-specific (ETS)-related gene (ERG) [9,10]. These endothelial cells in HCH do not stain for glucose transporter-1 (GLUT 1) (Figure 3) [5]. Genetic studies have revealed that HCH harbors mutations in GNAQ and GNA11, with a subset alternatively having mutations in PIK3CA [11]. These mutations are not seen in other vascular tumors and confirm HCH’s distinct biology.

## 4. Hepatic Infantile Hemangioma

Hepatic infantile hemangioma (HIH) is the most common benign liver tumor in infants and shows GLUT-1-immunopositivity., [12]. Three subtypes of HIH are reported, namely focal, multifocal, and diffuse [13]. Although HIH generally follows a benign clinical course, some tumors have been reported to undergo malignant transformation [8].

### 4.1. Clinical Presentation

There is no clarity regarding the natural history of HIH. Its clinical course is different to HCH, as it continues to proliferate until approximately 6 to 12 months old and then it gradually involutes until 3 to 9 years of life [4,12]. It is as twice as common in females as in males. Most HIH tumors are asymptomatic and remain undetected or are found incidentally upon post-natal imaging. However, some children may be symptomatic due to their size, location, or hemodynamic effects and can present with abdominal distension, a palpable right hypochondrial mass, abdominal compartment syndrome, anemia, and failure to thrive [14]. High-output cardiac failure may occur if arteriovenous or portovenous shunting is present. The infant may present with clinical hypothyroidism due to a high expression level of type 3 iodothyronine deiodinase in these vascular lesions, which inactivates the thyroid hormone. Laboratory findings have shown a low T4 and a high TSH [15].

### 4.2. Imaging Findings

Diffuse or multifocal HIH shows homogenous or centripetal enhancement, adjacent flow voids, enlargement of hepatic arteries or veins, and arteriovenous shunts upon imaging [15]. Diffuse lesions completely replace the liver parenchyma. Multifocal lesions are defined as multiple round tumors without complete parenchymal effacement. 

### 4.3. Pathology

Macroscopically, HIH shows white-tan nodules with occasionally degenerative changes in the centre. HIH is characterised by anastomozing, sinusoidal vascular channels (Figure 4) [13]. These channels are lined by a flattened, minimally, or moderately plump endothelium with small, bland, hyperchromatic nuclei without severe atypia; mitosis is infrequent. Redundancy features such as pseudopapillae formation, mimicking Masson papillary endothelial hyperplasia, are also reported, and some cases may show bilayering [8]. Lesional anastomozing channels are separated by variable amounts of fibrotic stroma. The periphery of the lesion shows active growth in the form of mitotic activity. Bile ducts and hepatocytes may be entrapped at the growing edge. HIH is not capsulated, with irregular margins. One of the hypotheses suggests hypoxia as an etiological factor for infantile hemangioma. Hypoxia-inducible factor-1-alpha (HIF-1α) leads to the activation of GLUT -1, vascular endothelial growth factor A (VEGF-A), and insulin-like growth factor 2 (IGF-2). Infantile hemangioma shows positivity for CD 34 and CD 133, just like embryonic veins, suggesting an arrest at an early stage of vascular differentiation [16].

## 5. Epithelioid Hemangioendothelioma

### 5.1. Clinical Presentation

Epithelioid hemangioendothelioma is an uncommon malignant tumor that is mostly reported in adults but can occasionally be seen in children [17]. In children, it usually presents as a multifocal disease with involvement of the liver, lungs, bone, and soft tissue [18]. The median age at diagnosis is 12 years and most cases are diagnosed in females [19]. 

### 5.2. Imaging Findings

Hepatic epithelioid hemangioendothelioma demonstrates a distinctive growth pattern upon CT and MRI characterized by the peripheral distribution of neoplastic lesions, subcapsular retraction, and a “lollipop sign”, which consists of a hepatic or portal vein tapering and terminating at the periphery of the tumor [18].

### 5.3. Pathology

Firm, ill-defined, and sometimes confluent nodules with infiltrative borders are identified upon macroscopic examination [18,20]. Histologically, cords and nests of dendritic or epithelioid cells in a variable fibromyxoid stroma are seen (Figure 5). The neoplastic dendritic cells are irregular and elongated or stellate with processes. Intracytoplasmic vacuoles can be seen. Erythrocytes can be seen in the intracytoplasmic vacuoles. The epithelioid cell types are round to polygonal and contain an eosinophilic cytoplasm, round nuclei, and inconspicuous nucleoli [18]. Epithelioid cells often grow as small clusters inside thin-walled vascular spaces. Tumor cells grow along sinusoids and eventually replace the intervening hepatocytes. Tumor cells also infiltrate portal structures and central veins. Immunohistochemical studies show positivity of endothelial markers including CD31, CD34, ERG, FLI-1, factor VIII, and lymphatic marker podoplanin (D2-40). Epithelioid hemangioendothelioma can also express cytokeratin [21]. Two distinct variants of epithelioid hemangioendothelioma have been recognized. One shows WWTR1–CAM1 fusion, is more common, and is seen in 90% of patients [22,23,24]. The other 5% of tumors show YAP1–TFE3 fusion [25]. These two variants show distinctive morphological features. Epithelioid hemangioendothelioma with WWTR1–CAMTA1 fusion shows neoplastic cells embedded in a myxohyaline stroma. These cells may be singly scattered or arranged in cords. It shows a sinusoidal growth pattern at its interphase with the normal liver. The native portal tract vessels show tumor cell plugging. Upon IHC, strong and diffuse nuclear expression of CAMTA-1 is identified in such cases [26]. Epithelioid hemangioendothelioma with YAP1–TFE3 fusion shows nests of endothelial cells with round vesicular nuclei, prominent nucleoli, and abundant glassy cytoplasms. Vasoformation is identified, which distinguishes it from WWTR1–CAM1 fusion-associated tumor. The IHC results show a loss of expression of the C-terminus of YAP-1, along with strong and diffuse nuclear positivity for TFE3 [25]. Epithelioid hemangioendothelioma with YAP1–TFE3 fusion has a higher chance of multifocal disease and metastasis as compared to tumors with WWTR1–CAM1 fusion; however, the former has a 5-year progression-free survival range of 85–90% as compared to the latter [27,28]. Tumors have been stratified into low-, intermediate-, and high-risk groups based on their mitotic activity, nuclear atypia, and presence of necrosis. A study by Shibayama et al., showed overall 5-year survival rates of 100%, 81.8%, and 16.9% in low-, intermediate-, and high-risk tumors, respectively [29].

## 6. Hepatic Angiosarcoma

Hepatic angiosarcoma is a highly malignant tumor and is exceedingly rare in children. It may be present anytime from neonatal age up to adolescents, with a median age of 40 months. A female preponderance is seen [30].

### 6.1. Clinical Presentation

Neonates and infants may present with abdominal distension due to hepatomegaly or with abdominal compartment syndrome [31]. In childhood, it presents as an abdominal mass along with abdominal pain, jaundice, emesis, or respiratory distress [30]. Laboratory findings may show anemia and consumptive coagulopathy. The serum AFP is normal, although elevated βHCG levels have been reported [31]. Hepatic angiosarcoma behaves aggressively and may show pulmonary metastasis at the time of initial diagnosis [31].

### 6.2. Imaging Findings

The ultrasound results may show a mixture of hyperechoic and hypoechoic areas due to hemorrhage and necrosis. CT shows a hypoattenuating mass as compared to the surrounding liver parenchyma and may show hyperattenuating areas representing foci of hemorrhage [32]. Using contrast-enhanced CT, it exhibits heterogeneous enhancement, which indicates central necrosis and fibrotic changes. Using MRI, angiosarcoma has a heterogeneous appearance due to areas of hemorrhage, necrosis, and fibrosis [33]. Using T2-weighted images, the mass appears hyperintense with hypointense internal septae. Using T1-weighted images, it appears as a low-intensity lesion with focal areas with increased signals due to hemorrhage.

### 6.3. Pathology

A large single tumor or multiple tumor nodules, either separate or coalescing, sometimes with infiltrative borders replacing the hepatic parenchyma, are noted upon macroscopic examination. The cut surface can be variegated with areas of hemorrhage and necrosis. Cystic spaces and a fleshy appearance are also known to occur [34,35]. The classic morphological features of angiosarcoma include markedly atypical endothelial cells with high nuclear-to-cytoplasmic ratios, hyperchromatic nuclei, inconspicuous to prominent nucleoli with solid spindled or epithelioid areas, and destructive invasion into the surrounding liver parenchyma and increased mitotic activity (more than the proliferative phase of HIH) (Figure 6) [8]. Hepatic angiosarcoma in children may arise within a non-involuting HIH, and in this context often shows GLUT-1 immunopositivity. Using IHC, the tumor cells are positive for CD31 and CD34 along with variable podoplanin positivity [34]. The IHC for CAMTA1 is negative [26]. MAPK pathway alterations are seen in 50% of angiosarcoma and TP53 or CDKN2A alterations in 20–30% [36].

## 7. Mesenchymal Hamartoma

Mesenchymal hamartoma (MH) is the second most common benign liver tumor in children, which usually presents before 2 years of age, with 15–20% patients presenting in the neonatal period. It has a slight male preponderance [37,38,39,40]. A definite pathogenesis is not known, although it is proposed that MH arises from the aberrant development of mesenchyme in the portal tracts during late embryogenesis [37,41].

### 7.1. Clinical Presentation

MH may be detected prenatally using ultrasound examination or may present with prenatal complications such as pre-term labor, maternal toxemia, fetal hypdrops, and fetal demise. MH has a prominent cystic component and fluid accumulation within the cyst may lead to rapid enlargement of the mass, leading to massive abdominal distension, diaphragmatic compression, and respiratory distress [38,39,42]. Inferior vena cava compression may lead to engorged veins over the anterior abdominal wall or lower limb edema [34]. Smaller lesions are asymptomatic or may present with non-specific symptoms such as fatigue, abdominal pain, and fever. They may even be detected incidentally upon imaging. The AFP levels may or may not be elevated.

### 7.2. Imaging Findings

The mass may be predominantly cystic with thick or thin septae or it may be solid with few cystic spaces. Thus, the imaging features may vary depending on the percentage of solid and cystic components. Using ultrasound, cystic MH is anechoic with thin or thick echogenic septa. Low-level echoes may be seen within the fluid, reflecting the gelatinous contents. Round hyperechoic parietal nodules within the cystic spaces are specific for MH. The solid component is isoechoic to the surrounding liver and shows minimal vascularity. The mixed solid cystic tumor shows a sieve-like appearance upon ultrasound with multiple anechoic cystic areas and intervening isoechoic to hyperechoic solid tissue [43,44,45,46]. Using CT, MH appears as a complex cystic mass and the cystic component of water attenuation is not enhanced after contrast administration. The stroma hypoattenuates as compared to the surrounding liver and the solid component and the septae are enhanced after contrast administration [43,47,48,49]. Using MRI, the solid component appears hypointense as compared to the adjacent liver parenchyma in both T1- and T2-weighted images due to fibrosis. The cystic portion shows a high signal intensity level in T2-weighted images and variable signal intensity in T1-weighted images based on the protein content in the fluid [47,48,49,50]. 

## 8. Pathology

MH may show great variations in size ranging from a few centimetres up to greater than 30 cm [51]. Typically, an un-encapsulated solitary lesion is reported; however, cases with multifocal growth have also been reported [51,52,53]. The cut surface shows solid and cystic components in varying proportions containing clear serous, mucoid, bile-tinged, or blood-tinged fluid with no necrosis or calcification (Figure 7). Larger lesions have more prominent cystic areas. The cystic structures do not communicate directly with the biliary tree. Histologically, MH shows a nodular growth pattern (Figure 8) with epithelial elements comprising branching bile ducts (resembling ductal plate malformation) admixed with haphazardly arranged bland spindle cells, embedded in mesenchymal stroma (Figure 9). The abnormal bile ducts are lined with cuboidal to columnar cells containing basally located round to oval nuclei and a lightly eosinophilic cytoplasm [54]. Occasionally, mucinous differentiation is also reported. The mesenchymal component is typically loose and myxoid in nature because of the accumulation of glycosaminoglycans. A thin collagenous band separates bile ducts from this loose mesenchyme (Figure 10). Relatively normal-appearing hepatocytes arranged in clusters or thin hepatic cell plates are identified (Figure 11). It has been argued that the intralesional hepatocytes are most likely to be derived from common progenitor cells and are a component of MH rather than entrapped non-neoplastic hepatocytes [54]. Extramedullary hematopoiesis is occasionally seen. The cystic spaces are lined by flattened to cuboidal epithelial cells surrounded by connective tissue. Pseudocysts with no epithelial lining are also seen. 

Hematoxylin and eosin staining alone is sufficient to provide a diagnosis of MH. The biliary epithelium and entrapped hepatocytes show immunoreactivity for pancytokeratin stains, while the pseudocysts and mesenchymal cells show vimentin expression. Immunoreactivity to smooth muscle actin, α-1-antitrypsin, and desmin is also reported in the mesenchymal component [51]. Recurrent chromosomal rearrangements involving 19q13.3/13.4, including t(11;19) (q11;q13.3/13.4) involving MALAT1 on chromosome 11, have been identified in MH [55,56,57]. The locus from chromosomes 19q13.3 to 13.4 harbors the largest known human micro-RNA cluster [58]. This chromosome 19 micro-RNA cluster represents an imprinted locus in which the expression of the maternal allele is silenced and the paternal allele is expressed during placental development [59]. These chromosomal rearrangements are found in the spindle cells but not in the bile ducts, suggesting that the admixed bile ducts are not neoplastic. A subset of MH harbors an unusual genetic alteration called androgenetic–biparental mosaicism, in which the tumor cells harbor two copies of the paternal allele instead of one copy each of the maternal and paternal alleles [60]. In this subset of cases, one allele is methylated and silenced while the other is demethylated and expressed. There are reports of MH in association with placental mesenchymal dysplasia, Beckwith–Wiedemann syndrome (BWS), and DICER1 alterations [51].

### 8.1. Inflammatory Myofibroblastic Tumors

Inflammatory myofibroblastic tumors (IMFTs) of the liver are uncommon mesenchymal neoplasms, mostly reported as case reports or small series in the literature [61,62]. IMFT is a distinctive tumor having intermediate malignant potential with a tendency for local recurrence and rare metastases and varying ages of presentation ranging from 3 months to 15 years [61]. There is no gender predilection [61]. Occasional case reports have described an early presentation in the neonatal period [63]. The cause of IMT is unknown, although cytogenetic changes suggest that these lesions may be of clonal origin [62].

### 8.2. Clinical Presentation

The patient can present with fever, abdominal pain, weight loss, abdominal mass, vomiting, or diarrhea. Jaundice caused by either the presence of a porta hepatis tumor or involvement of the hepatic hilum has also been reported [64,65,66]. The serum AFP level is normal. Inflammatory markers such as ESR and CRP are elevated.

### 8.3. Imaging Findings

The imaging features are non-specific and pose difficulties in providing a correct diagnosis. Using ultrasound, IMFT appears as an isoechoic or hypoechoic lesion [67]. The mass appears as a hypointense shadow when using CT and upon contrast administration it shows a peripheral enhancement in most cases, while some cases show a homogeneous enhancement [68]. Using MRI, both T1- and T2-weighted images show low signal intensity mass levels [69].

### 8.4. Pathology

IMFTs may be solitary or multiple, being well-circumscribed, firm, and white-tan, with a solid or whorled cut surface. Focal necrosis may occur. The IMFT is composed of sheets and loose fascicles of spindle cells admixed with inflammatory cells, embedded in a variably myxoid to collagenous stroma (Figure 12). The spindle cells have vesicular nuclei, prominent nucleoli, and a moderate cytoplasm and show evidence of myofibroblastic differentiation in variable proportions. The inflammatory cells comprise plasma cells, lymphocytes, histiocytes, eosinophils, and neutrophils (Figure 13). Three histologic growth patterns have been reported in the literature [70]. The myxoid–vascular pattern shows loose spindle cells in a myxoid and edematous stroma along with a network of small blood vessels. The compact spindle cell pattern shows a denser proliferation of spindle cells within an inflammatory background. There is a hypocellular fibrous pattern with extensive areas of collagenization resembling desmoid fibromatosis or scar tissue and relatively sparse inflammation. Using IHC, the spindle cells stain for smooth muscle actin (SMA) (Figure 14) and sometimes desmin. Approximately 15% of IMFTs express cytokeratin [70]. Furthermore, 50–60% of tumors show anaplastic lymphoma kinase (ALK) gene rearrangement (mostly translocation), resulting in the aberrant expression of the ALK chimeric protein upon IHC (Figure 15) [71]. A minor subset of tumors show the rearrangement of ROS1, ETV6, or NTRK3 as a possible oncogenic mechanism [72]. There is no apparent prognostic difference between ALK-positive and ALK-negative IMFTs [72].

## 9. Malignant Rhabdoid Tumor

The kidneys are the most common site for malignant rhabdoid tumor (MRT), with primary liver MRT being rare. It is a malignant tumor, with most patients presenting before 1 year of age. MRT of the liver is highly aggressive and approximately 60% of patients show metastasis at presentation and 90% of patients die of the disease [73,74].

### 9.1. Clinical Presentation

MRT of the liver presents with fever, abdominal distension, abdominal pain, anorexia, or vomiting. Upon examination, hepatomegaly or a right upper quadrant mass may be identified [75,76,77,78,79]. Using laboratory studies, there may be anemia, leukocytosis, elevated liver enzymes, and raised LDH levels. The AFP level may be normal or mildly elevated. Rarely, patients may present with spontaneous tumor rupture [80,81].

### 9.2. Imaging Features

The use of ultrasound shows a heterogeneous solid mass that may have cystic changes and calcification [76,77,79,82]. In cases of rupture, hematoma may be seen around the tumor. The use of contrast-enhanced CT reveals hypodense areas. The use of MRT shows heterogeneous hyperintensity upon T2-weighted MRI [83].

### 9.3. Pathology

Macroscopically hepatic MRT presents as a large lobulated mass often located in the right lobe with frequent areas of necrosis and hemorrhage. MRT shows sheets of asymmetrical, discohesive, undifferentiated, polygonal tumor cells with eccentrically placed vesicular nuclei, prominent nucleoli, and an abundant, brightly eosinophilic cytoplasm [53]. The classic rhabdoid morphology may be identified only focally; therefore, IHC may be important in confirming the diagnosis. MRT can show a small-cell growth pattern displaying round tumor cells with hyperchromatic nuclei and a scanty cytoplasm arranged in sheets (Figure 16). Rhabdoid tumor demonstrates a loss of expression of integrase interactor 1 (INI-1) in immunohistochemistry studies [84], leading to SMARCB1 inactivation. A loss of expression of INI-1 is identified in almost all hepatic cases of MRT [85,86]. INI-1 is a tumor suppressor gene involved in SWI/SNF chromatin remodeling complex. MRTs also display a heterogenous immunoprofile, with the expression of neural, mesenchymal, and epithelial markers. Most cases express vimentin, epithelial membrane antigen (EMA), or broad-spectrum cytokeratins. There may be variable immunoexpression of neuron-specific enolase (NSE), synaptophysin, glial fibrillary acidic protein (GFAP), and smooth muscle actin. Desmin and CD34 are rarely expressed [87]. MRTs also show a perimembranous staining pattern with CD99, a potential cause of confusion with Ewing’s sarcoma. Thus, INI1 mutation is the most crucial immunohistochemical and molecular test to be performed to identify malignant rhabdoid tumor.

## 10. Embryonal Sarcoma

Embryonal sarcoma of the liver (ESL) is an aggressive mesenchymal tumor accounting for 9–15% of all pediatric liver tumors and is the third most common childhood hepatic malignancy after hepatoblastoma and hepatocellular carcinoma [88,89,90,91,92,93]. It was first described by Stocker and Ishak in 1978 [94]. ESL usually occurs between 6 to 10 years of age and has no gender predilection. It has been reported to arise from MH, as these tumors share common genetic aberrations in chromosome 19q13 [95,96].

### 10.1. Clinical Presentation

The patient presents with abdominal pain, an abdominal mass or distension, weight loss, and vomiting [88,89,90,91,92]. Acute presentation may occur if the tumor ruptures due to rapid growth [89,97]. Metastasis to the lungs, lymph nodes, peritoneum, or pleura may occur [80,84]. The laboratory findings may reveal elevated liver enzyme levels, hypoalbuminemia, and anemia. Leukocytosis may occur in cases with intratumoral hemorrhage or necrosis and tumor rupture [89,92]. The serum AFP level is usually normal. False-positive ELISA results for parasites such as Echinococcus and Entamoeba histolytica have been reported due to molecular mimicry [98].

### 10.2. Imaging Findings

ESL is a solitary lesion that is more common in the right lobe of the liver [88]. Using ultrasound, ESL appears as a predominantly solid hyperechoic mass with focal cystic anechoic areas. CT scans on the other hand show a predominantly cystic, hypoattenuated mass because the gel-like myxoid stroma has a water-like density when using CT. This discrepancy of a predominantly solid mass upon ultrasound and cystic mass upon CT is diagnostic of embryonal sarcoma [99,100]. Another characteristic feature is the presence of serpiginous vessels in the tumor in CT scans. Using MRI, ESL appears as hyperintense in T2- and hypointense in T1-weighted images [88,89,100].

### 10.3. Pathology

The lesion is a large and well-circumscribed expansive mass with a fibrous pseudocapsule. The cut surface is solid and yellow-tan with cystic and myxoid or gelatinous areas. Foci of hemorrhage and necrosis may be seen [89,92]. Histologically, it shows markedly pleomorphic stellate- to spindle-shaped cells embedded in a myxoid to fibrous stroma. Giant tumor cells containing prominent cytoplasmic, PAS-positive hyaline globules are characteristic. The mitotic count is high, with frequent atypical mitosis and a high Ki 67 proliferation index value [88,92,97,100]. The diagnosis is based mainly on morphological features because of a non-specific immunophenotype. Desmin and cytokeratin expression is identified in 50% of cases [100,101]. ESL is known to harbor chromosome 19 alterations, just like MH, and both are considered to be components of the same biological spectrum; it is postulated that ESL may arise as a result of the malignant transformation of MH. The chromosome 19 micro-RNA possibly inhibits tumor suppressor genes and is implicated in tumorigenesis [102,103]. In addition to chromosome 19 alterations, other genetic changes implicated in the pathogenesis of ESL are TP 53 inactivation and complex copy number alterations [104].

## 11. Hepatobiliary Rhabdomyosarcoma

Hepatobiliary rhabdomyosarcoma (RMS) is a malignant tumor of myogenic origin that may present as an intraluminal biliary mass or cluster of grapelike masses [105,106]. Hepatobiliary RMS accounts for approximately 1% of all pediatric RMS cases, and it is the most common pediatric malignant tumor of the biliary tract [105,106].

### 11.1. Clinical Presentation

The tumor occurs mainly in children at a median age of 3 years, with a clear preponderance of boys. Hepatobiliary RMS typically presents with jaundice and obstruction of the bile duct, and occasionally cholangitis.

### 11.2. Imaging Findings

Radiological studies usually demonstrate a hypoechoic intraductal or periductal solid cystic lesion with dilation of an obstructed biliary tract.

### 11.3. Pathology

Embryonal RMS is variably cellular, ranging from stellate cells in an abundant myxoid stroma to sheets of poorly differentiated, densely packed rounded cells with round to slightly angulated hyperchromatic nuclei (Figure 17) [107]. Embryonal RMS protrudes into the lumen of the bile duct, imparting a “botyroid” appearance. Characteristically, there is a hypercellular layer, the “cambium layer”, just underneath the epithelium, which contains densely packed tumor cells. Mitoses may be seen but the tumor cells do not display the pleomorphism seen in undifferentiated embryonal sarcoma. Hepatobiliary RMS shows positive immunostaining for desmin, myogenin, and MyoD1 (Figure 18).

## 12. Epstein–Barr-Virus-Associated Smooth Muscle Tumor

Children with absent or severely impaired T-cell function, such as from primary immunodeficiency (PID), post-organ transplantation immunosuppression, or acquired immunodeficiency syndrome (AIDS), commonly develop Epstein–Barr virus (EBV)-induced lesions owing to uncontrolled proliferation of the infected cells [108,109,110]. While the majority of EBV-driven malignancies are lymphoid proliferations such as post-transplant lymphoproliferative disorder (PTLD) or AIDS-associated lymphomas, other histogenetically different neoplasms such as EBV-associated smooth muscle tumor (SMT) can also develop [111].

### 12.1. Clinical Features

The clinical signs of EBV-SMT are non-specific and mainly depend on the location, size, and organ displacement or disruption [112]. EBV-SMT can be unicentric or multicentric.

### 12.2. Imaging

There are no pathognomonic imaging features for EBV-SMT and a biopsy is essential for a formal diagnosis [108].

### 12.3. Pathology

Histologically, EBV-SMT is primarily composed of spindle to elongated neoplastic cells with an eosinophilic cytoplasm and indistinct cytoplasmic borders predominately arranged in interlacing fascicles resembling leiomyomas or leiomyosarcomas (Figure 19) [108,109,110]. The mitotic activity level is low, and necrosis and nuclear pleomorphism are uncommon. Immunohistochemistry studies have shown immunoreactivity to vimentin and smooth muscle markers such as SMA (Figure 20), caldesmon, muscle-specific actin, and desmin. The Epstein–Barr-encoding region (EBER) in situ hybridization (ISH) results show nuclear positivity in tumor cells (Figure 21). LMP1 protein expression has also been reported in some tumors.

## 13. Epithelial Tumors

### 13.1. Hepatocellular Adenoma

Hepatocellular adenomas (HCAs) are rare pediatric liver tumors. HCA is present in adolescents with a mean age of 14 years and comprises less than 5% of all pediatric liver tumors [113,114]. The pediatric patients do not show any gender predilection as is seen in adult patients, in whom female predominance is seen. Based on genotypic and phenotypic characteristics, HA cases were initially divided into 4 subtypes: hepatocyte nuclear factor 1 alpha (HNF1α)-inactivated HCAs (H-HCAs), inflammatory HCAs (I-HCAs), beta-catenin-mutated Has (b-HCAs), and unclassified HCAs (U-HCAs) [115]. A further evaluation using gene expression profiling, RNA sequencing, and whole-exome and -genome sequencing resulted in an expanded classification that includes b-HCAs involving exon 3 and exon 7 or 8, I-HCAs with beta-catenin mutations, and a newly diagnosed entity of sonic hedgehog HCAs (sh-HCAs) [116]. In a recent study on pediatric HCAs, b-HCA was the predominant subtype in the pre-pubescent group [114]. I-HCAs and H-HNF1α-mutated adenomas comprise the majority of tumors in the post-pubescent group.

### 13.2. Clinical Presentation

HCA is discovered incidentally upon imaging but patients may present with abdominal pain. Approximately 25% of patients across all age groups present with tumor rupture and intratumoral or intraperitoneal hemorrhage [117]. This is more common with the inflammatory subtype. Malignant transformation to hepatocellular carcinoma is rare in children and has an incidence rate of 4% across all age groups. The AFP levels are usually normal and a rise in AFP indicates malignant transformation [118]. An association of HCA has been reported with excessive estrogen exposure, norethindrone and norethisterone (synthetic progesterone) administration, and excess androgen exposure, and the tumor regresses after the factors causing excessive hormone exposure are alleviated [119,120,121,122,123,124]. HCA has an incidence range of 75–80% in patients with glycogen storage disease (GSD) and it develops during the second and third decades of life. Individuals with GSD are prone to having multiple HCAs [125]. Chromosomal aberrations involving a simultaneous gain of chromosome 6p and loss of chromosome 6q are seen in GSD patients developing HA [126]. A high frequency of β-catenin mutations is found in children with GSD harboring HA [127]. Patients with congenital extrahepatic portosystemic shunts are also prone to developing HCA, and it is proposed that excess oxygen delivery due to arterialization and increased estrogen and insulin delivery due to diversion of the splanchnic blood flow are the responsible factors [127]. Pediatric Has may also arise in the background of familial adenomatous polyposis syndrome, Fanconi anemia (FA), galactosemia, Hurler syndrome, Alagille syndrome, Abernethy malformation, biliary atresia, congenital hepatic fibrosis, immunodeficiency, cardiac hepatopathy status post-Fontan procedure, germline *HNF1A* mutations, and maturity-onset diabetes of the young type 3, among others [113,114,128]. HCAs may also occur spontaneously in pediatric settings.

### 13.3. Imaging Findings

The appearance of HCA on imaging is homogeneous and similar to the surrounding liver, but those with hemorrhage and fat show varied features [43]. Using ultrasound, HCA appears as a well-delineated, heterogeneous solid mass. Tumors with a high lipid content or recent hemorrhage appear hyperechoic to the surrounding liver parenchyma. In cases of an old hemorrhage, the lesion appears hypoechoic, similar to a cyst. Ultrasonography is used as a screening tool in patients with GSD and annual screening is usually recommended [129]. CT scans show a spherical, sharply delineated mass that appears to be hypoattenuating compared to the surrounding liver, while areas of recent hemorrhage appear to be hyperattenuating, thereby giving a heterogeneous appearance [129,130,131,132]. Using contrast-enhanced CT, HA shows a heterogeneous enhancement during the arterial phase while hyperattenuating compared to the surrounding liver. In the portal venous and delayed phases, it becomes isoattenuating or presents with rapid wash out [43,133]. Using MRI, HA is heterogeneous and predominantly hyperintense in T1- and T2-weighted images [124]. After gadolinium contrast administration, HA shows an early arterial enhancement and then becomes isointense to the surrounding liver in portal venous and delayed phase images. An enhanced pseudocapsule may be visible via delayed acquisition [134].

### 13.4. Pathology

Macroscopically, HA appears as a well-demarcated lesion with thin or no fibrous capsule [135]. Histologically, it shows thin liver cell plates composed of normal-sized hepatocytes along with numerous arteries unaccompanied by bile ducts or a portal vein. The cytoplasm may be eosinophilic, clear, or may show fat. Mitosis and significant nuclear atypia are not seen. Intratumoral hemorrhage or fibrosis may be seen [132]. HA with HNF-1α inactivation shows prominent intralesional steatosis (Figure 22), and upon IHC shows decreased or absent immunostaining for liver fatty acid-binding protein (LFABP). Furthermore, β-catenin-activated HA shows pseudo-acinar formation and cholestasis with patchy cytologic atypia (Figure 23). Using IHC, this tumor shows nuclear positivity for β-catenin (Figure 24) and diffuse positivity for glutamine synthetase (Figure 25). CTNNB1 mutation is detected in these tumors. Inflammatory HA is characterized by inflammatory cell infiltration, dystrophic arteries, sinusoidal dilatation and congestion, and the presence of a ductular reaction at the periphery of the lesion (Figure 26) [115]. Further immunostaining results of these tumors’ reactivity for CRP (Figure 27) and serum amyloid A (SAA) (Figure 28) are noted. The IL6ST mutation is identified in these tumors.

## 14. Focal Nodular Hyperplasia

Focal nodular hyperplasia (FNH) comprises 2–4% of all pediatric liver tumors, with an incidence rate of 0.02% in children [136,137,138]. Approximately two-thirds of children with FNH are females, with a median age of 8.7 years [139].

### 14.1. Clinical Presentation

Most FNH cases are detected incidentally during imaging and the patients are usually asymptomatic [140]. Around 20–36% children are symptomatic and present with a palpable abdominal mass and distension with or without abdominal pain. If the mass is large, it can cause portal hypertension and compression of the surrounding structures [141]. The serum AFP level is usually normal. Children with a history of solid tumors such as Wilm’s tumor and neuroblastoma and those with hematopoietic stem cell transplant (HSCT) are at an increased risk of developing FNH, with an incidence range of 5–12% [142]. These cancer survivors develop FNH 4–12 years after treatment and those with HSCT develop FNH earlier. Such patients are usually asymptomatic and FNH is detected during routine screening. These patients may have multifocal FNH, and their lesions are less likely to have central scars [143]. It is proposed that FNH occurs as a hyperplastic response of hepatic parenchyma to local arterial hyperperfusion and hyperoxygenation [144]. Children with extrahepatic congenital portosystemic shunts and biliary atresia are at risk of developing FNH [145,146]. Vascular alterations are responsible for the development of FNH in such patients.

### 14.2. Imaging Findings

Using ultrasound, FNH appears as a homogeneous, well-circumscribed mass with variable echogenicity, central or eccentric vascular supply, and a central stellate hyperechoic scar [43,147]. In 50% of cases, color Doppler images show an increased blood flow in the central scar extending to the periphery in a spoke–wheel pattern, with the flow being predominantly arterial. Using CT scans, FNH appears as a homogeneous, well-circumscribed mass that is isodense or slightly hypodense to the surrounding liver parenchyma and has a central scar that is hypoattenuating. Upon contrast administration, the lesion enhances homogeneously in the arterial and early portal venous phases. Due to its arterial supply, the enhancement is earlier and more intense than the surrounding liver parenchyma, and in the late portal venous and delayed phases it becomes Isoattenuating to the surrounding liver [43,148,149]. Using MRI, FNH appears homogeneous and isointense to slightly hypointense to the surrounding liver in T1-weighted images and isointense to slightly hyperintense in T2-weighted images in 75% of cases due to edema in the scar tissue. The mass enhances homogeneously after gadolinium administration. It appears hyperintense to the surrounding liver in the arterial phase. The mass may be slightly hyperintense or isointense to the surrounding liver in the portal venous phase. The central scar is enhanced during the delayed phase. The presence of a hyperintense T2-weighted central scar with delayed enhancement distinguishes FNH from fibrolamellar carcinoma, which shows a hypointense central scar in T2-weighted images and is not enhanced in delayed images [43,150,151]. The use of MRI with hepatobiliary-specific contrast agents helps to distinguish FNH from other liver lesions. FNH contains normal hepatocytes that readily take up the hepatobiliary-specific contrast agent, although it has malformed bile ducts that fail to excrete it, meaning the arterial enhancement is there for an extended period long after the other liver lesions have washed out [152,153].

### 14.3. Pathology

Grossly, FNH is a well-circumscribed lesion with a central scar in 70% of cases [146]. Central scars are uncommon in small lesions, although as the lesions become larger, a scar is noted frequently, generally being present in lesions greater than 4 cm. Microscopically, it is composed of cytologically bland hepatocytes arranged in one- to two-cell-thick plates with interspersed bands of fibrosis that contain bile ductules and focally thick-walled arteries (Figure 29). The hepatocytes are arranged in incomplete to complete nodules surrounded by bands of fibrosis. The hepatocytes are polyclonal and the reticulin framework is intact. The central scar is composed of mature collagen with numerous medium to large thick-walled arteries showing fibromuscular hyperplasia, myointimal proliferation, and myxomatous changes with luminal narrowing [154,155]. The fibrous septae also show variable inflammation and mild to prominent bile ductular reaction. The hepatocytes adjacent to the fibrosis septa can show cholate stasis with copper deposition, resulting from cholestatic injury due to a lack of normal bile ducts in FNHs. Portal tracts may be noted at the periphery of the lesion. There is an overexpression of glutamine synthetase, and on IHC it shows a map-like positivity in FNH (Figure 30) as compared to the normal liver, where the expression is limited to hepatocytes surrounding the central vein. The Ki-67 proliferation index is low and is similar to the background liver. The immunostaining results for glypican 3 and beta-catenin (nuclear staining) are negative. The SAA and CRP immunohistochemistry results can show patchy staining in FNH but should not have the strong and diffuse staining that is characteristic of I-HCA [146].

## 15. Hepatoblastoma

Hepatoblastoma (HB) is the most common primary malignant liver tumor in children [2]. The incidence rate of HB is slowly increasing, with a current range of 1.2–1.5 cases/million population/year [156]. HB usually occurs in children between 6 months and 3 years of age, with a slight male predominance and a male/female ratio of 1.6:1 [2,157,158]. Congenital HB has been reported and comprises less than 10% of pediatric HB cases [159]. There has been an increase in the incidence rate of HB, which is attributed to an increase in premature births, low birth weight survival rates, and earlier detection due to better imaging modalities [160]. It is postulated that increased oxidative damage due to various factors such as radiation, total parenteral nutrition, antibiotics, and oxygen therapy may increase the likelihood of the development of HB in premature and low birth weight children [161]. Most HB cases are sporadic and occur without any background liver disease. Some cases are associated with genetic abnormalities such as Beckwith–Wiedemann syndrome, familial adenomatous polyposis, trisomy 18, and hemihypertrophy [162].

### 15.1. Clinical Presentation

Children with HB present with abdominal pain, distension, and a palpable abdominal mass, along with other non-specific complaints such as anorexia, fatigue, and weight loss [162]. The presentation is acute in neonates, who may present with respiratory distress and decompensation in addition to an abdominal mass. The congenital HB may rupture during delivery, leading to perinatal hemorrhage and shock [163,164]. Fractures in the ribs and spine have been reported in children diagnosed with hepatoblastoma, and these children present with irritability and bone pain [165]. There are case reports of precocious puberty in infants with hepatoblastoma secondary to β-hCG secretion [166]. The AFP level is usually elevated in children with HB, except for the small-cell type, where the cells are not differentiated and do not secrete AFP. The patients may have thrombocytosis secondary to thrombopoietin produced by the tumor [167]. An improved outcome has been reported with multidisciplinary treatment [168].

### 15.2. Imaging Findings

Using ultrasound, HB appears as a lobular, well-circumscribed hypoechoic or heterogeneous lesion. Calcification, necrosis, and vascular invasion may be seen. Congenital HB may be detected using prenatal ultrasound, and such pregnancies may be complicated by polyhydramnios secondary to gastrointestinal compression by the mass, resulting in fetal hydrops [169]. Using CT scans, HB is heterogeneous and hypoattenuating to the surrounding liver parenchyma [170]. Using contrast administration, the enhancement is less than that of the surrounding liver; however, peripheral arterial enhancement is seen. Using MRI, HB is heterogeneous with hyperintense lesions in T2-weighted images and hypointense in T1-weighted images. Using contrast-enhanced MRI, HB appears as hypointense in the arterial, portal venous, and delayed phases [171,172].

### 15.3. Pathology

A macroscopic examination is important in hepatoblastoma. Hepatoblastomas usually involve the right lobe and present as a solitary tumor but can be multifocal. The cut surface is usually lobulated and may be whitish (Figure 31), yellow, brown, green, or variegated (Figure 32), depending on the differentiation of the tumor and whether a mesenchymal component is present [173]. Resections or explants specimens after neoadjuvant chemotherapy demonstrate a heterogeneous appearance with areas of hemorrhage and necrosis [174]. Calcification and ossification may be present. The tumor should be mapped to assess a post-neoadjuvant chemotherapy response and the morphology of the residual tumor, and to study its correlation with the prognosis. The non-neoplastic liver is usually normal.

Hepatoblastoma is classified broadly into epithelial and mixed epithelial and mesenchymal subtypes, and their histopathological features are summarized in Table 2. Epithelial-type HB is further categorized into fetal, embryonic, pleomorphic, macrotrabecular, cholangioblastic, and small-cell undifferentiated types [162,173,174,175,176]. Fetal HB is composed of cells that resemble fetal hepatoblasts during embryonic development. It is further categorized into (1) fetal cases with low mitotic activity (<2/10HPF) and (2) crowded fetal cases, which are mitotically active (≥2/10HPF). Fetal HB is composed of uniform polygonal cells arranged in 1–2-cell-thick trabeculae. There is no necrosis or pleomorphism. The tumor has classic light and dark areas based on cells with a clear or eosinophilic cytoplasm that show some degree of zonation (Figure 33). Extramedullary hematopoiesis may be seen (Figure 34). A diagnosis of well-differentiated fetal HB can be made only in a primary resection specimen where the tumor is entirely of the fetal type [176]. This diagnosis cannot be made with biopsy or post-chemotherapy specimens.

Crowded fetal HB shows closely packed cells, and the light and dark areas seen with the well-differentiated histology are not evident. The increased nucleocytoplasmic ratio gives this subtype a more crowded appearance (Figure 35). Nuclear pleomorphisms and atypical mitosis are absent, and the presence of these features suggest a pleomorphic fetal HB. Embryonal HB resembles the developing liver at 6–8 weeks of gestation. The cells have a high nucleocytoplasmic ratio, a scant cytoplasm with indistinct cell borders, and a large angulated to oval nucleus (rather than the round nucleus seen in the fetal subtype) with prominent nucleoli (Figure 36). Mitosis is frequent and necrosis may be seen. Pleomorphic HB is uncommon and is seen after chemotherapy, and neoplastic cells resemble mitotically active fetal HB or embryonal HB but show pleomorphism and increased and abnormal mitoses (Figure 37).

The small-cell undifferentiated (SCUD) subtype accounts for ~5% of all HB cases [177]. Historically, the SCUD subtype has an aggressive biology and worse survival rates [177,178,179]. Both INI-1positive and INI-1 negative SCUD HB cases are reported [176]. Tumors with a diffuse SCUD morphology are now classified as malignant rhabdoid tumors. SCUD HB with loss of INI-1 expression shows variations, from tightly packed small round cells in sheets with a scanty cytoplasm to a frankly rhabdoid phenotype with an eccentric nucleus, prominent nucleolus, and eosinophilic cytoplasm. SCUD HB with preserved INI 1 shows small cells with a high nucleocytoplasmic ratio, pale vesicular to hyperchromatic nuclei, scanty cytoplasm, and indistinct cell borders (Figure 38). The SCUD foci are mixed with an embryonal pattern. One recent study has shown that the presence of the SCU subtype in HB does not appear to adversely affect the patient outcome, suggesting that we should be able to treat patients with SCU HB according to risk stratification without regard for the presence of the SCU histology [180]. Small-cell components should be differentiated from the blastemal areas, which have a more ovoid to spindled morphology. Blastemal cells may be associated with mesenchymal elements such as osteoids. The blastemal cells may persist after chemotherapy; however, the SCUD areas are inconspicuous after chemotherapy. Macrotrabecular HB accounts for 5% of all HB cases and shows thick trabeculae, which may be more than 20 cells thick and may show the fetal, embryonic, or pleomorphic subtype of cells (Figure 39) [162]. They show strong nuclear β-catenin expression, in contrast with membranous β-catenin staining seen in pediatric hepatocellular carcinoma. Cholangioblastic HB shows a prominent ductular differentiation with nuclear β-catenin staining, which differentiates it from reactive ductular proliferation. Post-therapy-induced morphological changes in HB may be challenging for a pathologist [174]. Post-treatment tumors may show areas of tumoral “maturation”, including cytologic and architectural differentiation resulting in mimicry of the non-neoplastic liver (Figure 40). A morphology resembling HCC displaying nuclear anaplasia is reported. Dispersed, round cysts containing erythrocytes or hemosiderin-laden macrophages are identified in such specimens (Figure 41).

The mixed epithelial and mesenchymal type is subdivided into two categories based on the presence or absence of teratoid features. The most common mesenchymal element is osteoids (Figure 42) followed by cartilage. Spindle cells with cartilage and bone may be seen. The mesenchymal component is an integral part of the tumor and is not a result of chemotherapy or metaplasia. Teratoid HB shows mature glial elements, a primitive neuroepithelium forming tubules and rosettes, a melanin and retinal pigment epithelium, a squamous epithelium, and mucinous glands (Figure 43), along with blastemal elements. Rarely, neuroendocrine differentiation is also reported (Figure 44). IHC is quite helpful for classifying various types of HB. Glutamine synthetase shows diffuse cytoplasmic staining results (Figure 45) and glypican 3 shows fine granular cytoplasmic positivity (Figure 46) in the fetal morphology. The crowded fetal, pleomorphic fetal, and embryonal subtypes show coarse glypican 3 positivity (Figure 47) [162]. The SCUD foci show losses of hep-par 1 (Figure 48) and glypican 3 (Figure 49) and positivity for β-catenin (Figure 50) and INI-1 (Figure 51). The mesenchymal component shows nuclear β-catenin staining (Figure 52). The mesenchymal elements are negative for glypican 3. The neuroendocrine elements, showing positivity for INSM1 (Figure 53). HB is characterized by aberrant Wnt pathway activation that is most often associated with genetic alterations in CTNNB1 [181]. CTNNB1 mutations (encoding b-catenin) can be found in over 80% of HB cases. In 5% to 10% of HB cases, there is mutation in the NFE2L2 gene, and such cases are associated with a poor prognosis.

## 16. Pediatric Hepatocellular Carcinoma

Hepatocellular carcinoma (HCC) is the second most common primary liver cancer in children after HB [182]. Pediatric HCC presents in the age range of 10–14 years and is associated with cirrhosis in about one-third of cases [183]. In children, HCC develops in two distinct clinical settings—one with an underlying genetic or metabolic condition such as hemochromatosis, bile salt export pump deficiency, hereditary tyrosinemia type 1, GSD type 1, Alagille syndrome, or neurofibromatosis [184,185,186]. There are also reports that describe occurrence of HCC in multidrug resistance 3 protein (MDR 3) protein deficiency, tight junction protein 2 (TJP2) deficiency, and transaldolase (TALDO) deficiency [187,188,189,190]. The second type occurs in children without an underlying chronic liver disease. Hepatitis B virus is an important etiology for HCC, especially in endemic areas. With increasing immunization against hepatitis B, the demographic profile of pediatric HCC is changing, even in the endemic areas, with an increasing proportion of younger children having genetic and metabolic diseases presenting with HCC [184]. Biliary atresia is associated with HCC, and the prevalence rate is estimated to be 1.3% [190,191]. The identification of HCC is difficult in patients who have undergone a Kasai procedure, as such patients often have dominant regenerative nodules that mimic HCC. The serial monitoring of AFP levels and ultrasonography at regular intervals is recommended in such patients to look for new lesions [191].

### 16.1. Clinical Presentation

HCC presents as an abdominal mass and pain. In the late stages, there may be jaundice and cachexia [192]. Patients with cirrhosis can present with signs of decompensated liver disease and portal hypertension such as ascites, splenomegaly, variceal bleeding, and encephalopathy. About one-third of pediatric HCC patients are asymptomatic and the tumor is detected incidentally [192,193]. Fibrolamellar HCC may be associated with paraneoplastic syndrome, with androgen aromatization leading to gynecomastia and the production of thyroid hormone and β HCG by the tumor [194]. AFP is usually elevated and is useful for screening patients with cirrhosis or other predisposing conditions. Higher AFP levels are associated with higher fatality rates [195]. Patients with fibrolamellar HCC do not show elevated AFP levels.

### 16.2. Imaging Findings

Using ultrasound, the lesion appears as a heterogeneous hyperechoic mass with increased vascularity [196]. Using contrast-enhanced CT or MRI, the mass shows intense enhancement in the arterial phase and wash out in the portal venous and delayed venous phases [197]. Using MRI, HCC appears as a heterogeneous, hypointense neoplasm in T1-weighted images and mildly hyperintense to the surrounding liver in T2-weighted images. Finding a mass measuring more than 2 cm via CT or MRI that is large, hypervascularized, and with the wash out phenomenon has a 95% positive predictive value [198]. CT scan is the preferred modality to look for the extent of the tumor, resectability, presence of vascular invasion, and metastasis [198,199]. Fibrolamellar HCC needs to be differentiated form FNH, as in MRI results both show subtle deviations of signal intensity as compared to the surrounding liver parenchyma in pre-contrast T1- and T2-weighted images, with the presence of a central scar and arterial hyperenhancement. Fibrolamellar HCC, however, demonstrates greater heterogeneity due to areas of hemorrhage and necrosis. Fibrolamellar HCC has a hypointense central scar in T2-weighted images as compared to the hyperintense T2-weighted central scar seen in FNH. Fibrolamellar HCC demonstrates portal venous enhancement [200].

### 16.3. Pathology

Several histological subtypes of HCC have been reported in adults [201]. In pediatric patients, only a few HCC subtypes have been described. The recent WHO classification of pediatric liver tumors distinguishes fibrolamellar and non-fibrolamellar HCC cases (including conventional and clear-cell variants) [202,203]. The conventional morphology has been reported in 73% and the clear-cell subtype in 2% of HCC cases in children [204]. A scirrhous growth pattern has also been reported [205]. Our review of pediatric HCC identified conventional HCC with a trabecular architecture (Figure 54) and the clear-cell subtype of HCC [184,185,187]. Rare reports of pediatric HCC with syncytial giant cells (Figure 55) are available [190]. Conventional HCC cases are classified into well-differentiated, moderately differentiated, and poorly differentiated types [176]. Table 3 describes the morphological features of conventional HCC.

Fibrolamellar HCC accounts for 1% of all HCC and is more common in children and adolescents as compared to adults [206,207]. Grossly, it is a well-circumscribed, hard scirrhous mass with a central scar. Morphologically, neoplastic cells are monotonous, large, and polygonal with an abundant and deeply eosinophilic cytoplasm, centrally placed nuclei, vesicular chromatin, and prominent nucleoli (Figure 56). Bi- or multinucleation can also be seen. Architecturally, tumor cells show a trabecular pattern or clusters separated by dense bands of intratumoral fibrosis arranged in lamellar (parallel) arrays or an irregular pattern. A sheet-like growth pattern with minimal or no fibrosis is also reported [208]. Other morphological features of FL-HCC include the presence of pale bodies and hyaline bodies, which are present in about 50% of cases. The pale bodies are amphophilic, round, intracytoplasmic inclusions and are believed to be a mixture of acute-phase reactants, including fibrinogen. The composition of the hyaline bodies is unclear, although it is generally thought that they represent a similar phenomenon to pale bodies. The tumor cells may show a pseudoacini formation and a few cases may show mucin production. The background liver shows no significant inflammation or fibrosis. Fibrolamellar HCC shows immunohistochemical positivity for the histiocytic marker CD68 (Figure 57) and biliary marker CK-7 (Figure 58) [208]. Markers of hepatic differentiation (Hep-par1, arginase 1, and albumin mRNA, as detected via in situ hybridization) are also positive. The immunohistochemistry of liver fatty acid-binding protein (LFABP) is negative or weakly positive in FL-HCC [209]. The immunomarker anterior gradient 2 has also been reported to be positive in 75% of fibrolamellar HCC cases. In 2014, Honeyman et al. reported a recurrent DNAJB1-PRKACA fusion transcript in FL-HCC that occurred because of a specific microdeletion on chromosome 19, resulting in the upregulation of PRKACA activity by a promoter switch mechanism [210]. More than 95% of fibrolamellar HCC cases show a DNAJB1-PRKACA genetic alteration. Rare cases show biallelic mutations in PRKAR1A [211]. Fluorescence in situ hybridization (FISH) or polymerase chain reaction (PCR) can be utilized to identify the DNAJB1-PRKACA fusion gene. Fibrolamellar HCC cases have a unique transcriptomic signature characterized by the strong expression of specific neuroendocrine genes, including PCSK1, NTS, DNER, and CALCA, suggesting that these tumors may have a cellular origin different from that of HCC [212].

## 17. Hepatoblastoma with Carcinoma Features

Hepatoblastoma with carcinoma features (HBC) is a newer entity that displays histological and molecular characteristics that are a combination of HB and HCC. HBC is a genetically unstable tumor with high mutation rates and a poor outcome [213]. In 2014, the first International Consensus Classification of Pediatric Liver Neoplasms introduced a category of hepatocellular neoplasm not otherwise specified (HCN NOS) for tumors with combined and overlapping features of HB and HCC. [175].

HBC includes 3 histologic categories of tumors:Biphasic HCN NOS, which shows distinct areas with HB- and HCC-like features;Equivocal HCN NOS, with features intermediate between HB and HCC;Hepatoblastoma with a focal macrotrabecular pattern, pleomorphism, and anaplasia (HBFPA).

The second category, i.e., an equivocal hepatocellular neoplasm, resembles HB but shows a macrotrabecular pattern, acinar pattern, high N/C ratio, pleomorphism, anaplasia, a high mitotic count, and has giant tumor cells. HBFPA may focally show pleomorphic cells with large irregular hyperchromatic nuclei, intranuclear inclusions, prominent nucleoli, and atypical mitosis. Sumazin et al. accomplished the molecular profiling of theses tumors and found Wnt signaling pathway gene alterations (also seen in HB) in all patients with HBC. Further, 57% of patients showed alterations in pathways associated with stem cell pluripotency, including PI3K-AKT or mTOR signaling, as also seen in HCC. Mutations in genes associated with aggressive cancers, such as TERT, FGFR4, and KMT2C, were seen in 51%, 13%, and 10% of patients, respectively.

Table 4 summarizes age, gender preponderance, risk factor, and prognosis data for pediatric liver tumors.

## 18. Conclusions

Benign and malignant pediatric liver tumors show a wide variety of morphologic features and pose a significant challenge to a surgical pathologist. There is a continuous evolution in the pathologic classification of pediatric liver tumors as we update our understanding via molecular characterization and the correlation with its morphology. The proper utilization of immunohistochemistry markers helps in the accurate characterization of these tumors in daily practice. The use of international, multicenter, collaborative efforts and consensus workshops holds promise for furthering our understanding of these uncommon tumors. We hope that the pathological description and use of ancillary techniques described in this review will help pathology trainees, fellows, and practicing pathologists navigate through this group of diagnostically challenging tumors.

## Figures and Tables

**Figure 1 diagnostics-13-03524-f001:**
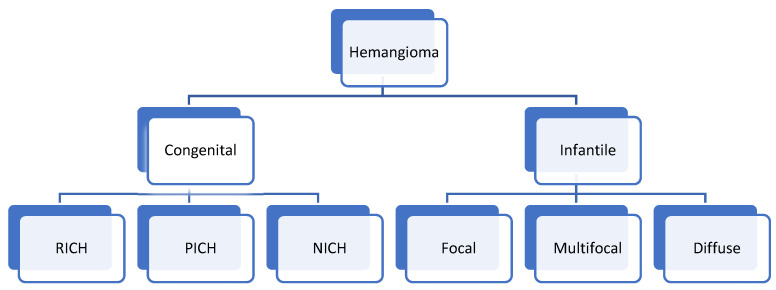
Types of hepatic hemangioma in children. RICH: Rapidly involuting congenital hemangioma; PICH: partially involuting congenital hemangioma; NICH: non-involuting congenital hemangioma.

**Figure 2 diagnostics-13-03524-f002:**
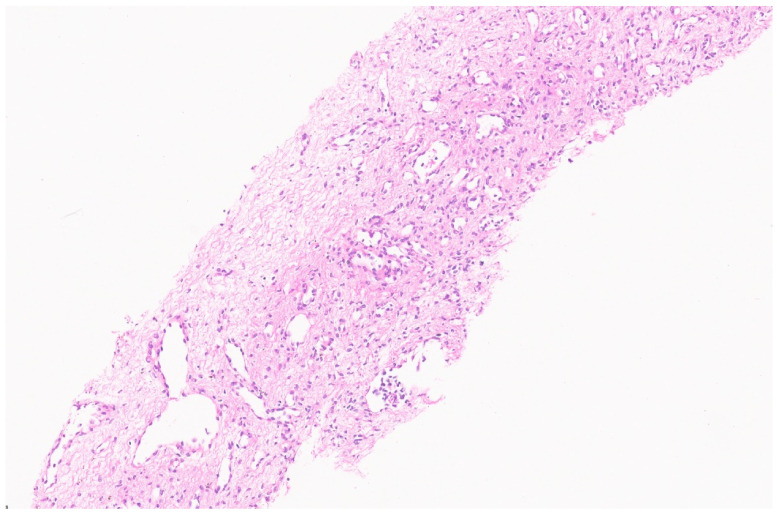
Hepatic congenital hemangioma with variously sized vascular channels (H&E, ×3).

**Figure 3 diagnostics-13-03524-f003:**
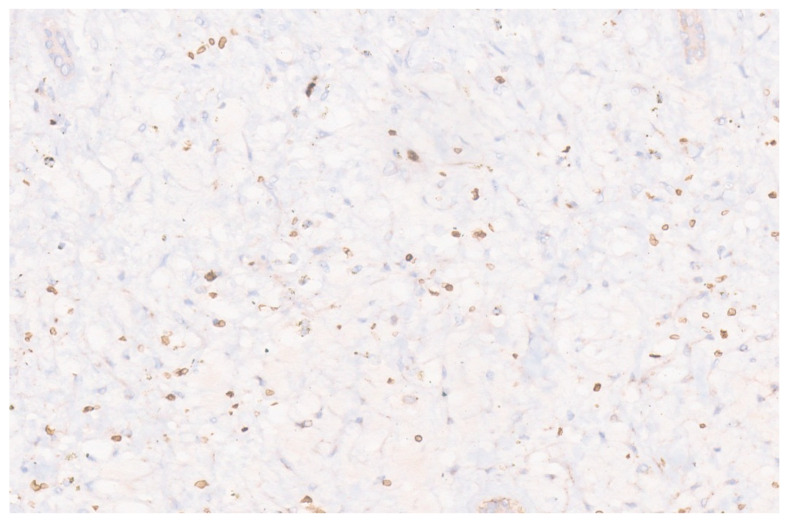
GLUT1 is negative in the endothelial cells. Red blood cells are the positive internal control (×15).

**Figure 4 diagnostics-13-03524-f004:**
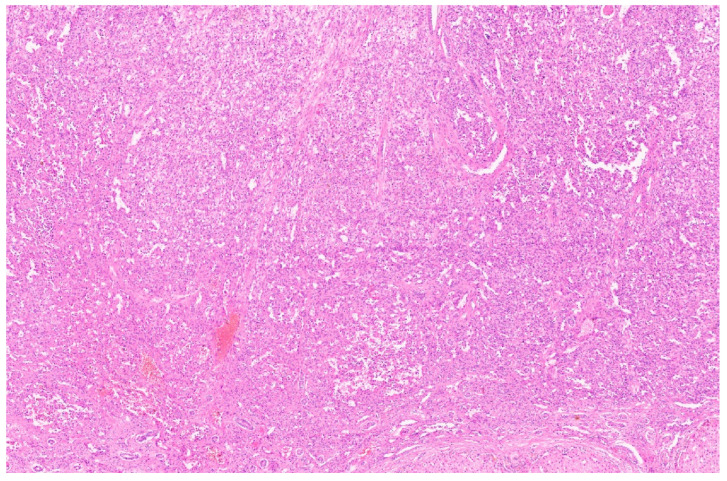
Hepatic infantile hemangioma with anastomozing vascular channels (H&E, ×3).

**Figure 5 diagnostics-13-03524-f005:**
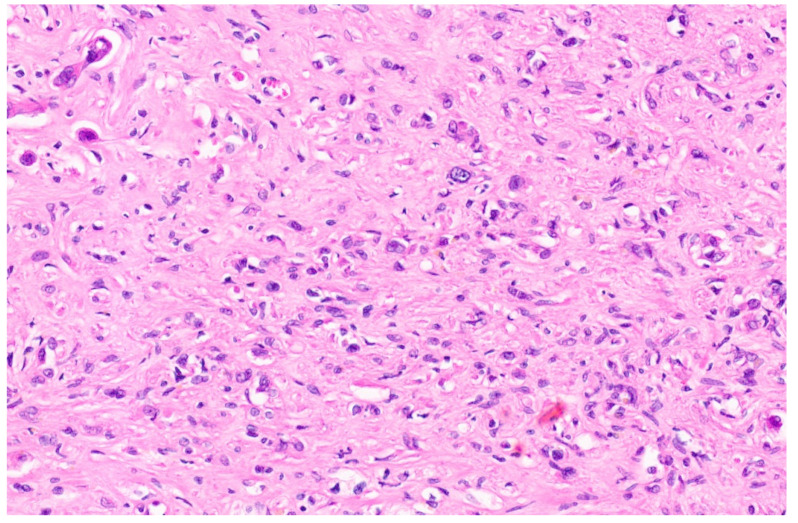
Hepatic epithelioid hemangioendothelioma with tumor cells embedded in fibromyxoid stroma (H&E, ×20).

**Figure 6 diagnostics-13-03524-f006:**
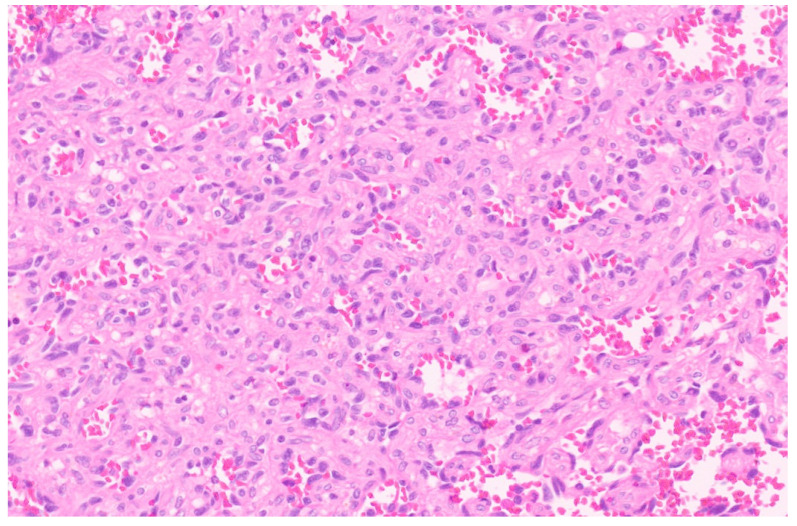
Hepatic angiosarcoma with malignant endothelial cells (H&E, ×20).

**Figure 7 diagnostics-13-03524-f007:**
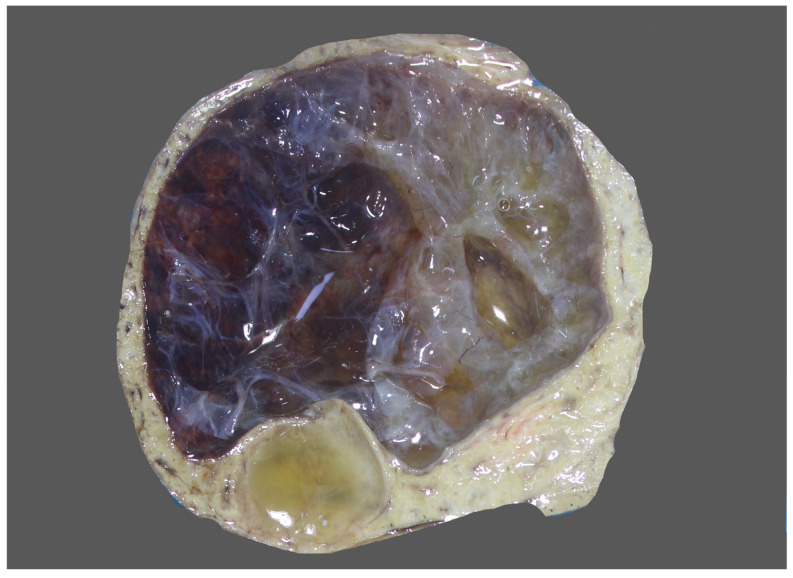
Mesenchymal hamartoma with marked cystic changes.

**Figure 8 diagnostics-13-03524-f008:**
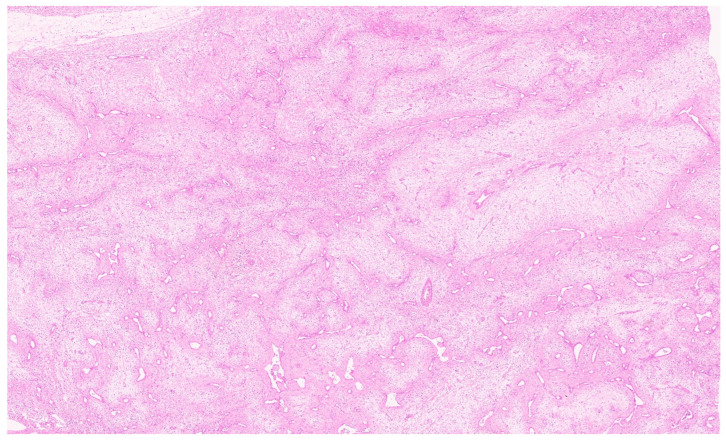
Mesenchymal hamartoma with branching bile duct structures and stroma (H&E, ×1).

**Figure 9 diagnostics-13-03524-f009:**
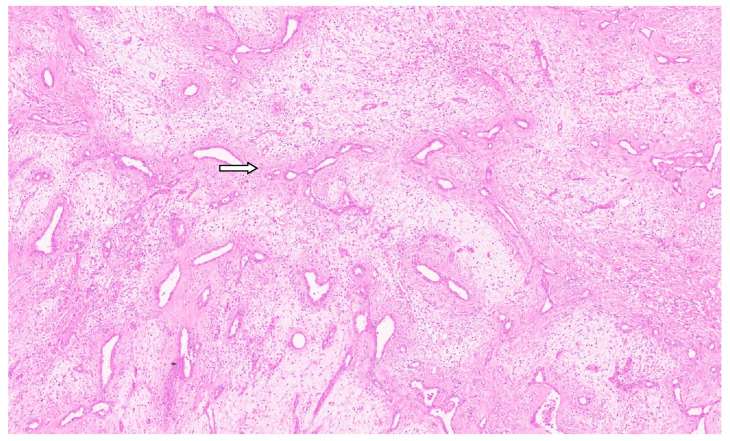
Mesenchymal hamartoma with branching ductal structures resembling ductal plate malformation (arrow, H&E, ×4).

**Figure 10 diagnostics-13-03524-f010:**
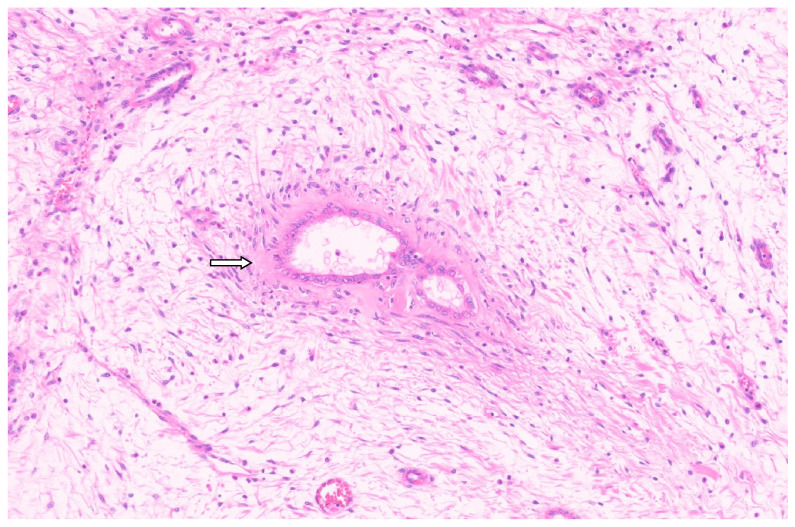
Collagenous band around the Bile ductss separating them from the loose mesenchyme (arrow for collagen band, H&E, ×11).

**Figure 11 diagnostics-13-03524-f011:**
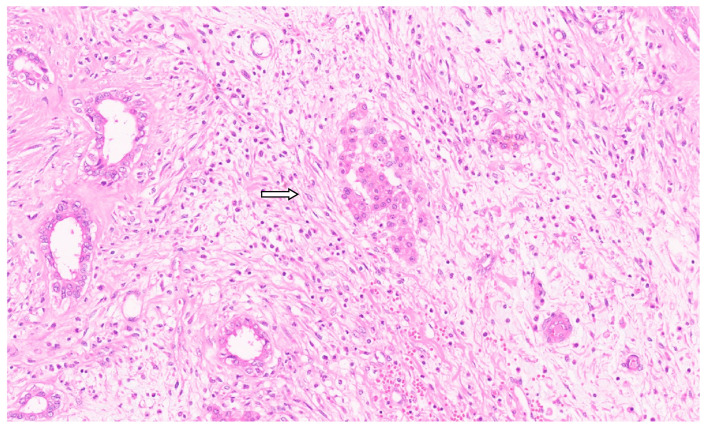
Entrapped hepatocytes in the tumor (arrow, H&E, ×12).

**Figure 12 diagnostics-13-03524-f012:**
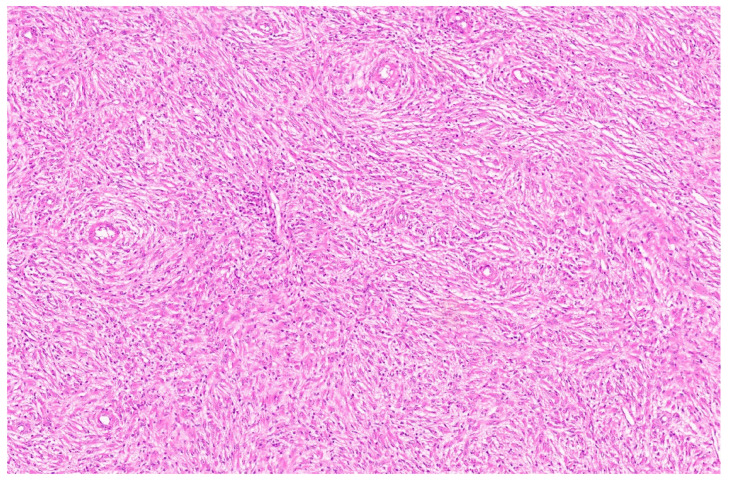
Spindle-shaped cells with interspersed inflammatory cells (H&E, ×5).

**Figure 13 diagnostics-13-03524-f013:**
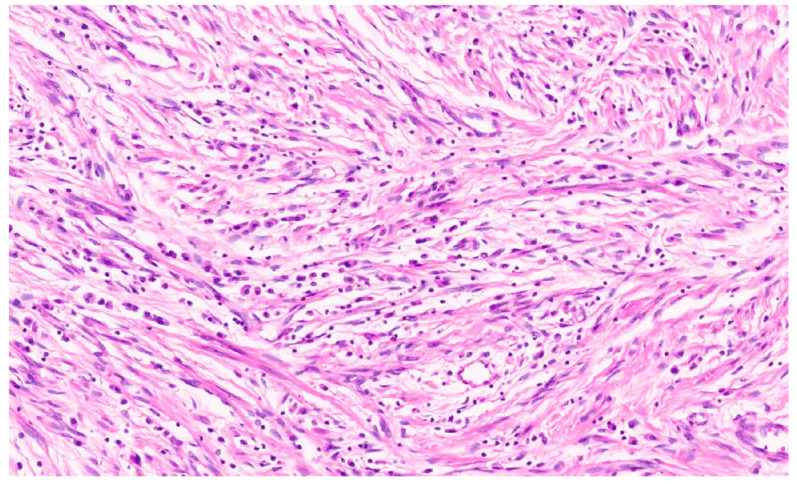
Inflammatory cells comprising lymphocytes, plasma cells, and histiocytes (H&E, ×18).

**Figure 14 diagnostics-13-03524-f014:**
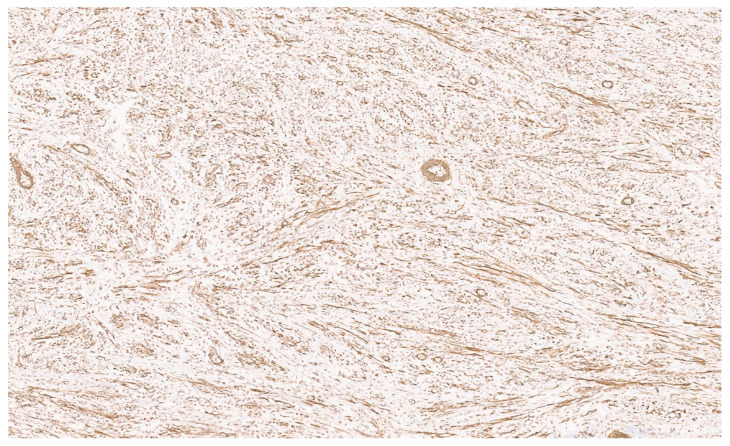
SMA immunopositivity in a liver IMFT (×6).

**Figure 15 diagnostics-13-03524-f015:**
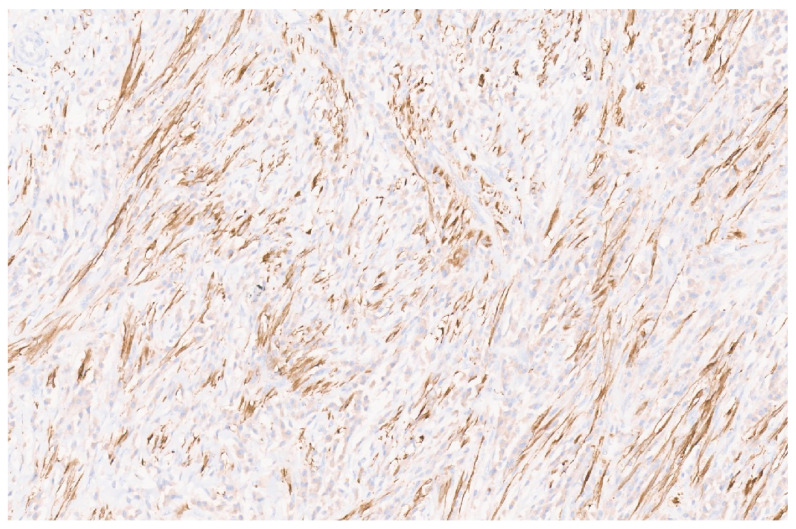
ALK immunopositivity in a liver IMFT (×10).

**Figure 16 diagnostics-13-03524-f016:**
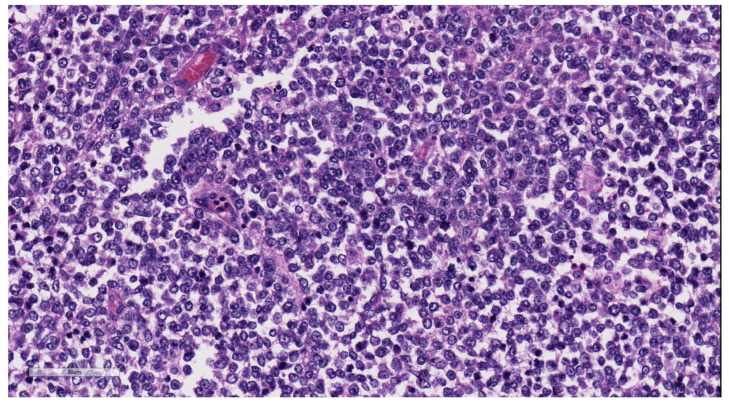
Rhabdoid tumor in a child with sheets of small cells (H&E, ×15).

**Figure 17 diagnostics-13-03524-f017:**
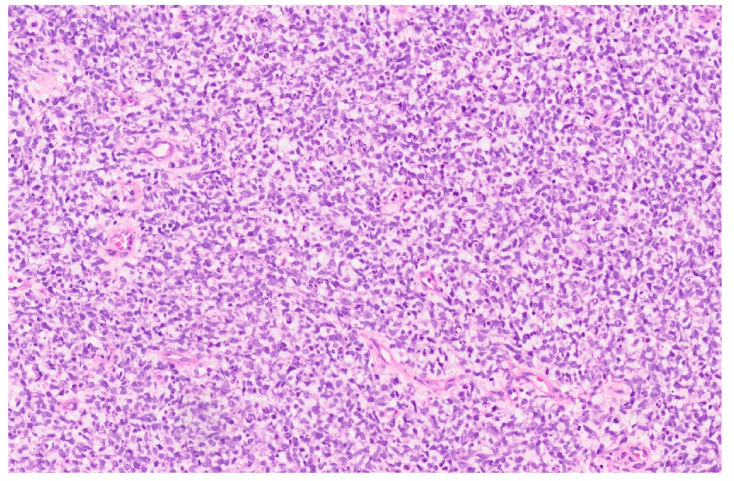
Embryonal rhabdomyosarcoma with round tumor cells in a myxoid background (H&E, ×5).

**Figure 18 diagnostics-13-03524-f018:**
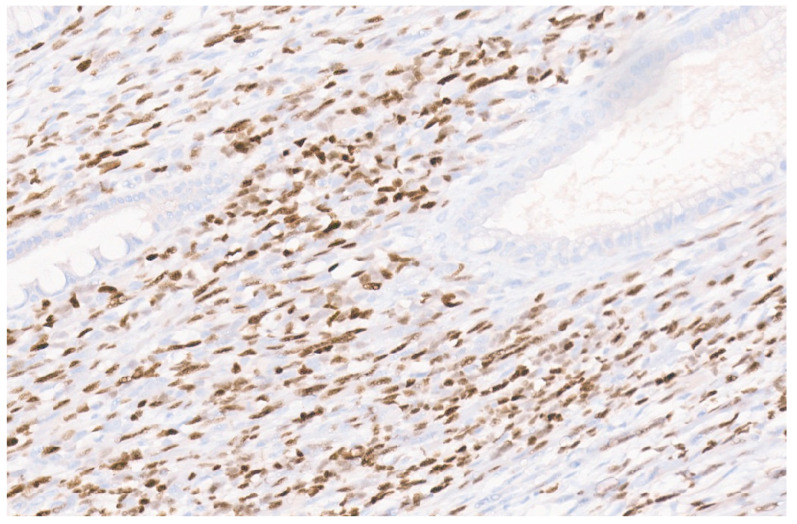
MYOD1 positivity in a hepatobiliary rhabdomyosarcoma (×20).

**Figure 19 diagnostics-13-03524-f019:**
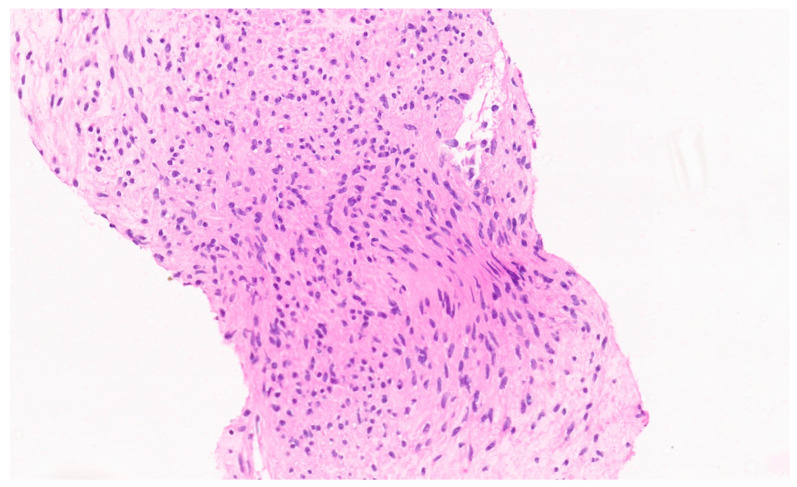
Epstein–Barr-virus-associated smooth muscle tumor (H&E, ×20).

**Figure 20 diagnostics-13-03524-f020:**
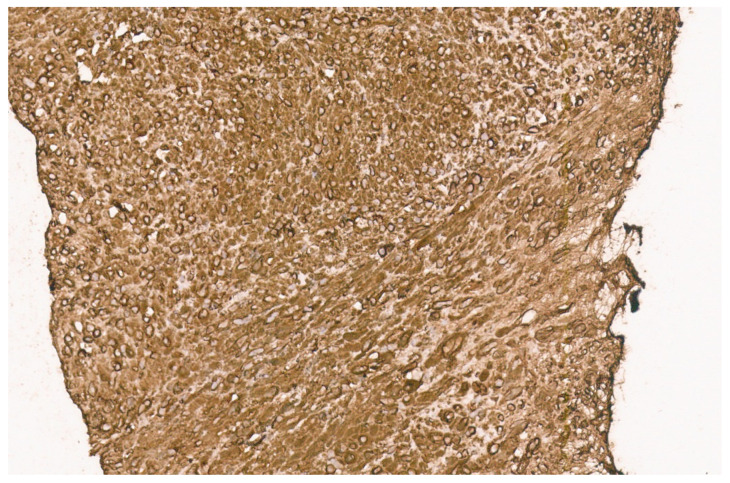
Strong SMA immunopositivity in the tumor cells (×20).

**Figure 21 diagnostics-13-03524-f021:**
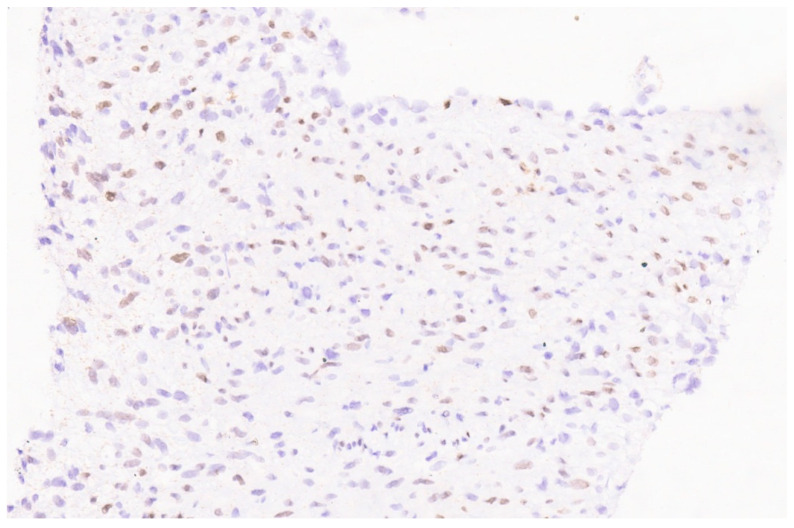
Epstein–Barr-encoding region (EBER) in situ hybridization (ISH) showing patchy nuclear positivity in tumor cells (×10).

**Figure 22 diagnostics-13-03524-f022:**
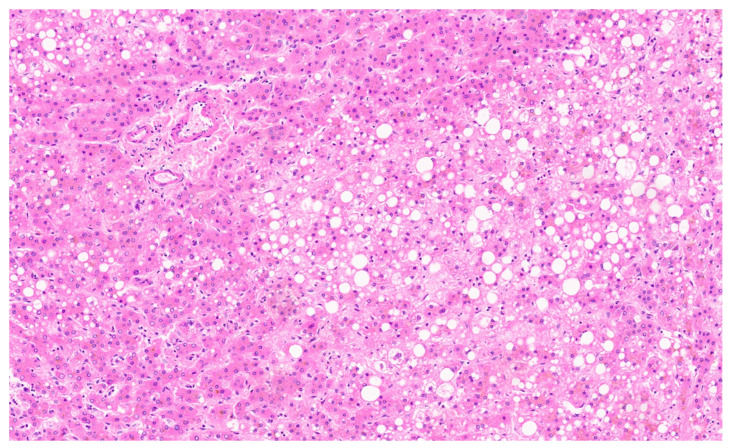
HNF-1α inactivated adenoma with thin liver cell plates composed of normal-sized hepatocytes with fat and numerous arteries (H&E, ×11).

**Figure 23 diagnostics-13-03524-f023:**
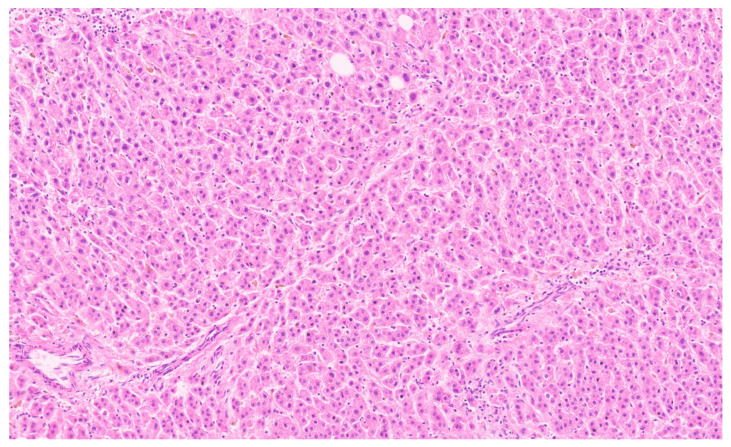
A β-catenin activated adenoma (H&E, ×7).

**Figure 24 diagnostics-13-03524-f024:**
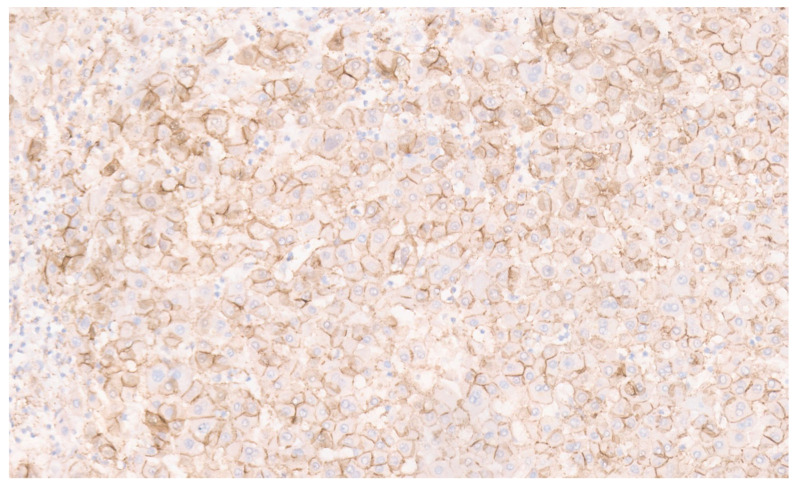
Patchy nuclear positivity for β-catenin (×10).

**Figure 25 diagnostics-13-03524-f025:**
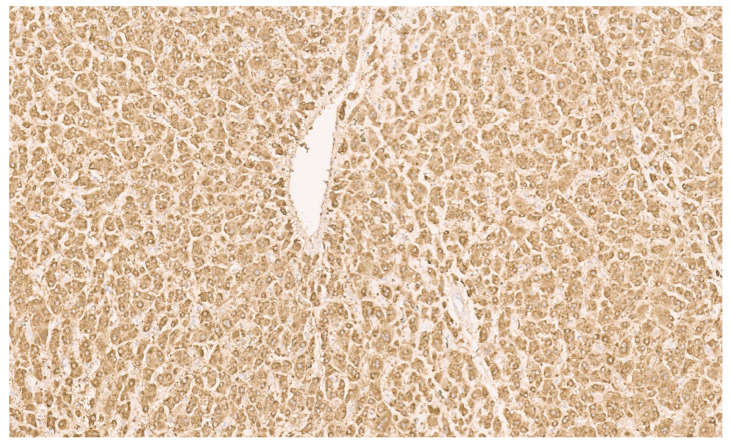
Strong glutamine synthetase positivity for β-catenin (×10).

**Figure 26 diagnostics-13-03524-f026:**
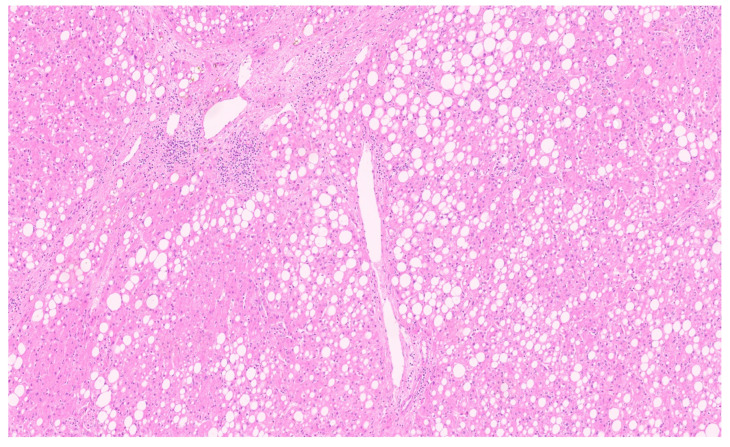
Inflammatory HCA with patchy inflammation and steatosis ((H&E, ×7).

**Figure 27 diagnostics-13-03524-f027:**
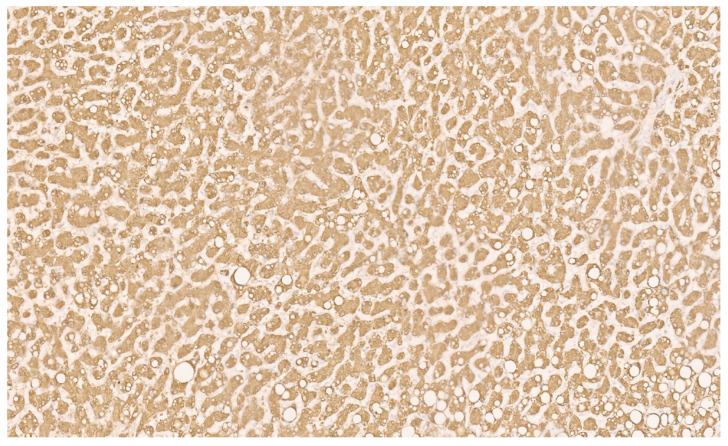
Inflammatory HCA with strong CRP immunostaining (×8).

**Figure 28 diagnostics-13-03524-f028:**
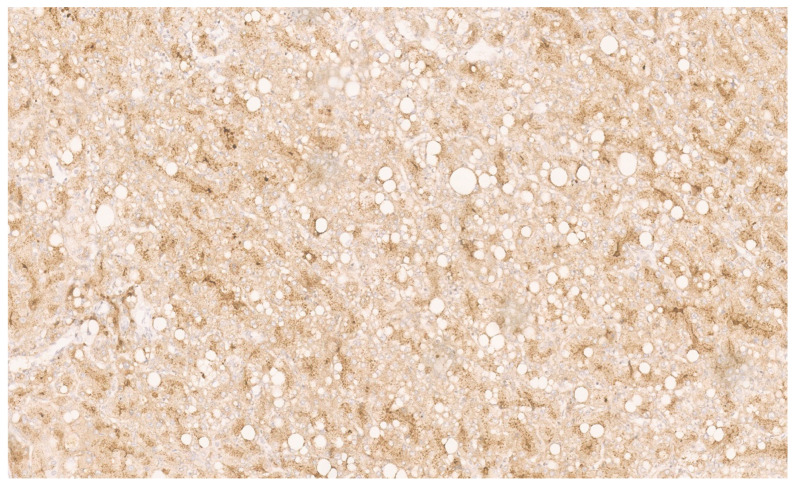
Inflammatory HCA with SAA immunostaining (×10).

**Figure 29 diagnostics-13-03524-f029:**
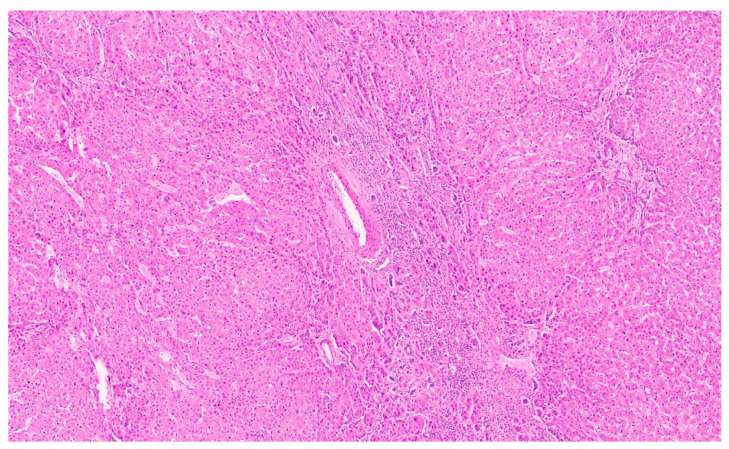
Focal nodular hyperplasia with cytologically bland hepatocytes with interspersed bands of fibrosis that contain bile ductules and arteries [H&E, ×10].

**Figure 30 diagnostics-13-03524-f030:**
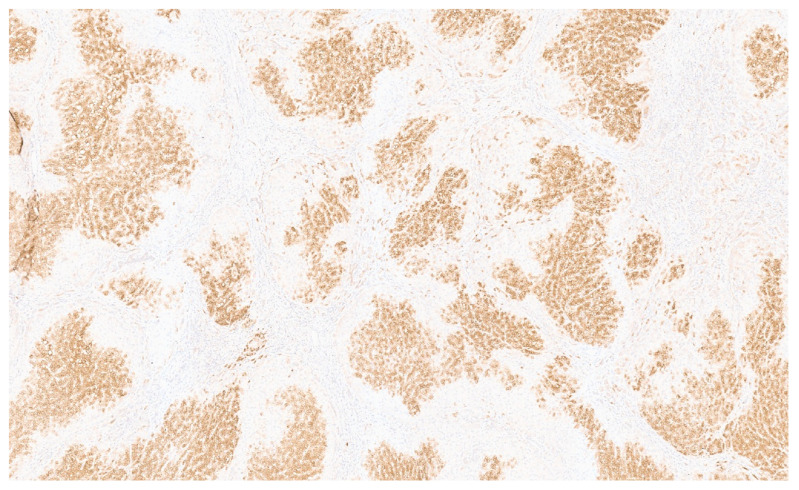
Glutamine synthetase with map-like positivity in FNH (×10).

**Figure 31 diagnostics-13-03524-f031:**
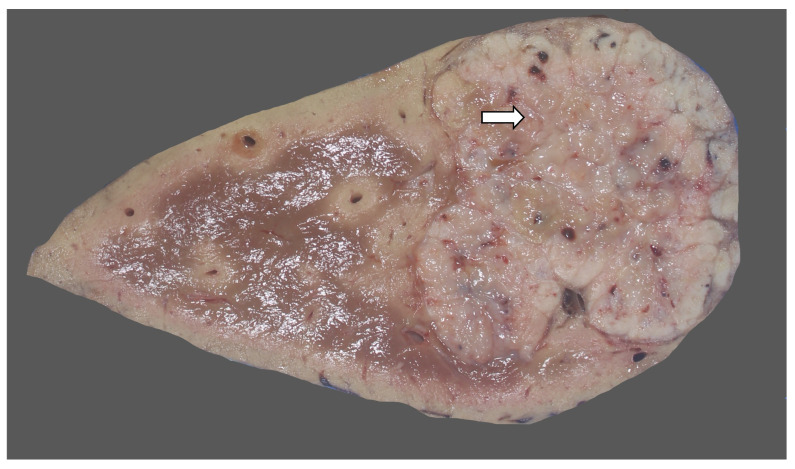
Hepatoblastoma with an epithelial component (arrow) and a few dark hemorrhagic foci. The non-tumoral liver is unremarkable.

**Figure 32 diagnostics-13-03524-f032:**
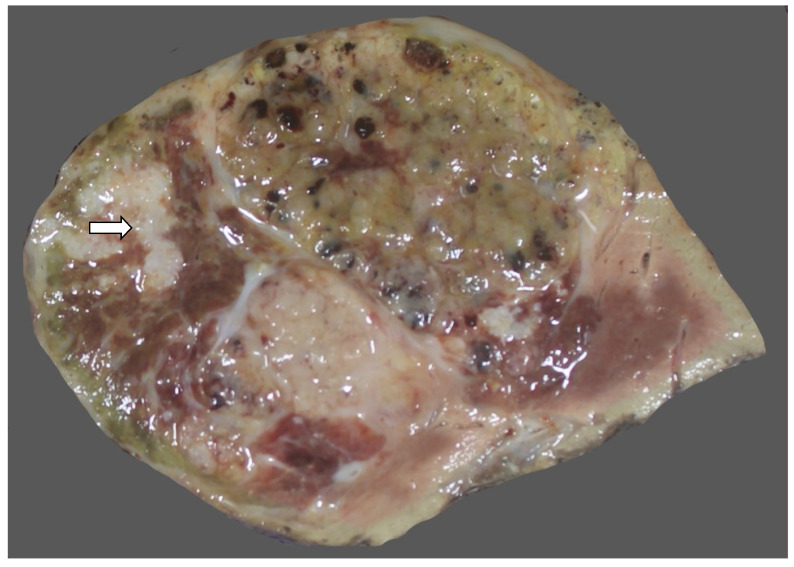
Macroscopy of mixed hepatoblastoma post-chemotherapy. The cut surface shows a whitish mesenchymal component (arrow) admixed with an epithelial component and dark hemorrhagic foci.

**Figure 33 diagnostics-13-03524-f033:**
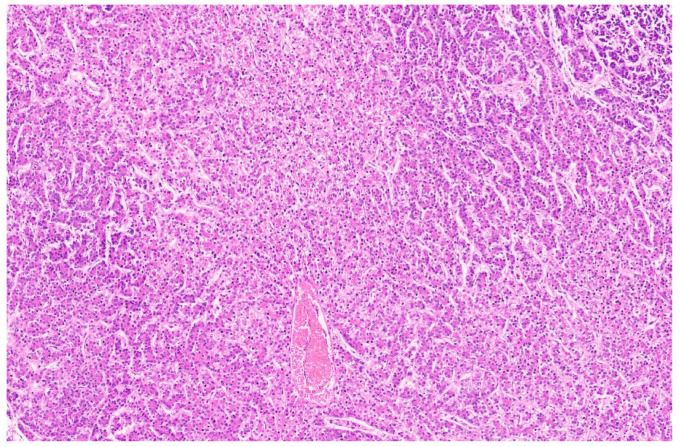
Well differentiated fetal HB displaying zonation with clear and eosinophilic areas (H&E, ×5).

**Figure 34 diagnostics-13-03524-f034:**
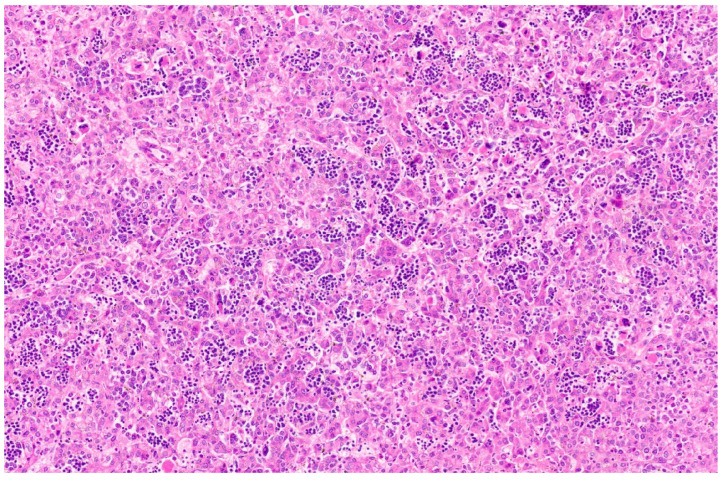
Fetal HB displaying extramedullary hematopoiesis (H&E, ×10).

**Figure 35 diagnostics-13-03524-f035:**
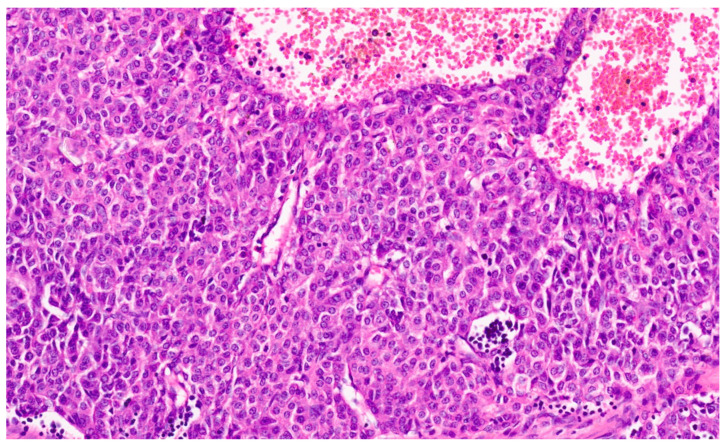
Crowded fetal HB displaying mitosis (H&E, ×15).

**Figure 36 diagnostics-13-03524-f036:**
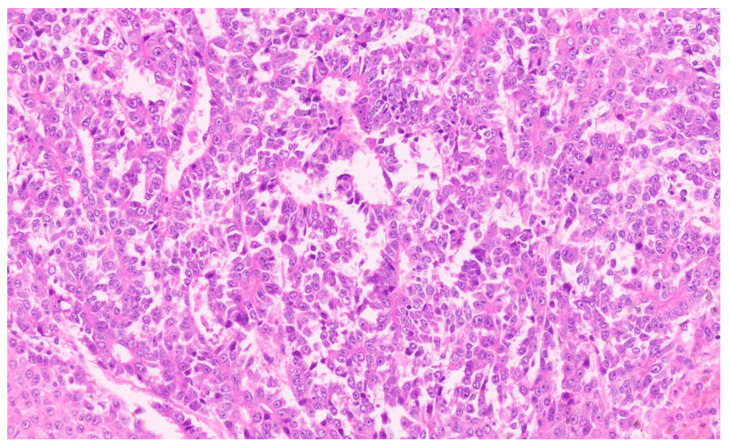
Embryonal HB with hyperchromatic nuclei and mitosis (H&E, ×12).

**Figure 37 diagnostics-13-03524-f037:**
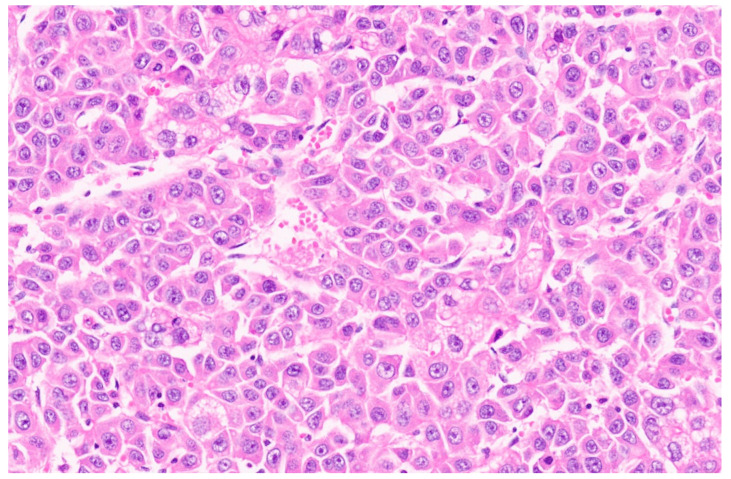
Hepatoblastoma with pleomorphic cells post-chemotherapy (H&E, ×20).

**Figure 38 diagnostics-13-03524-f038:**
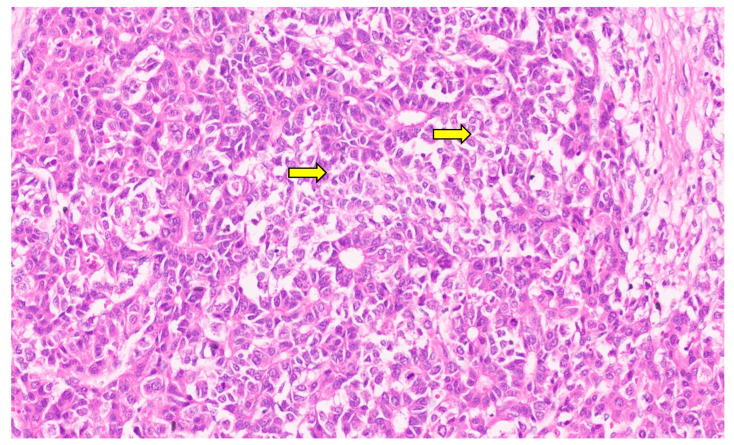
Small-cell undifferentiated foci (arrow) with embryonal HB (H&E, ×12).

**Figure 39 diagnostics-13-03524-f039:**
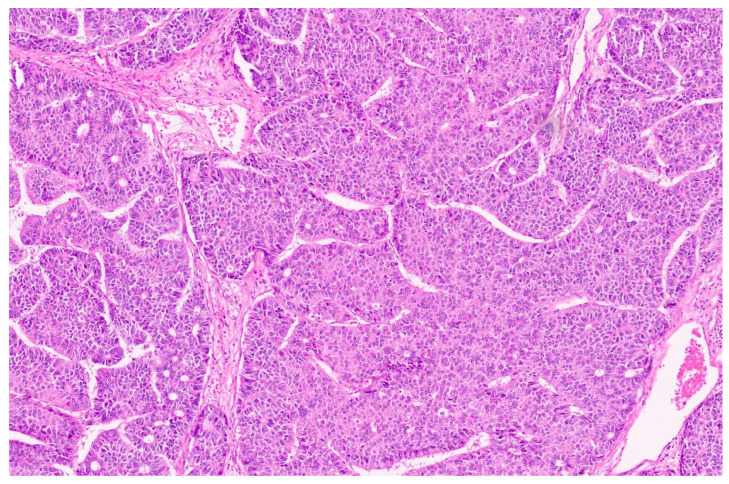
Macrotrabecular pattern in a hepatoblastoma (H&E, ×6).

**Figure 40 diagnostics-13-03524-f040:**
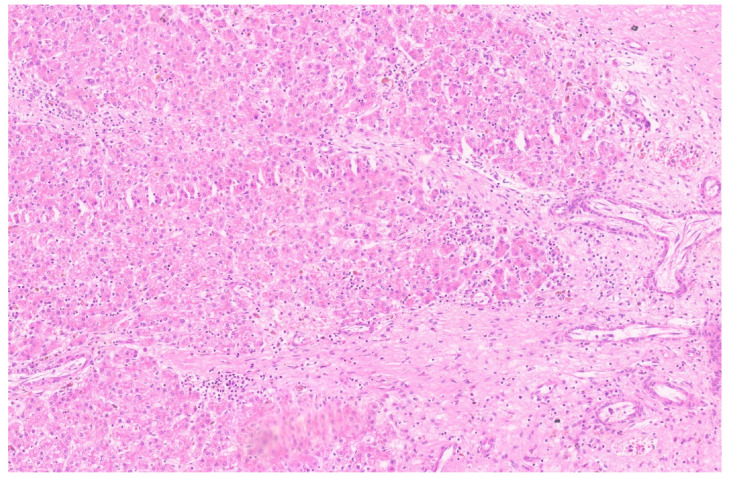
Cytoarchitectural “maturation” in HB mimicking the non-neoplastic liver. [×10].

**Figure 41 diagnostics-13-03524-f041:**
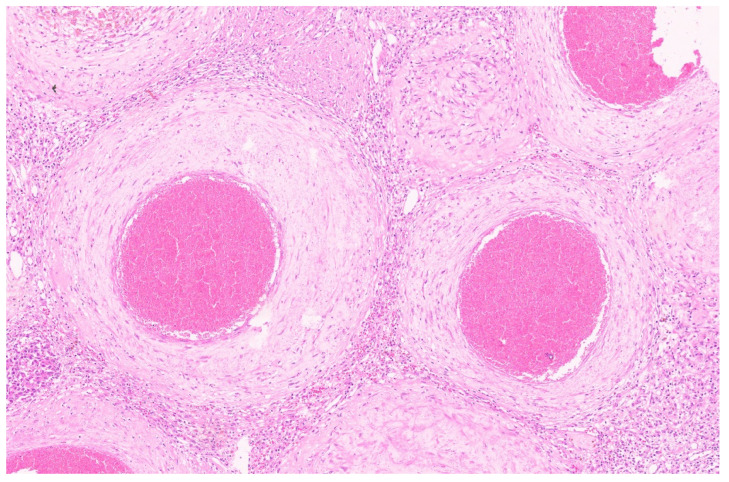
Peliotic-like foci characterized by round contours and uniform fibrous walls and containing packed erythrocytes in a hepatoblastoma post-chemotherapy. [H&E, ×15].

**Figure 42 diagnostics-13-03524-f042:**
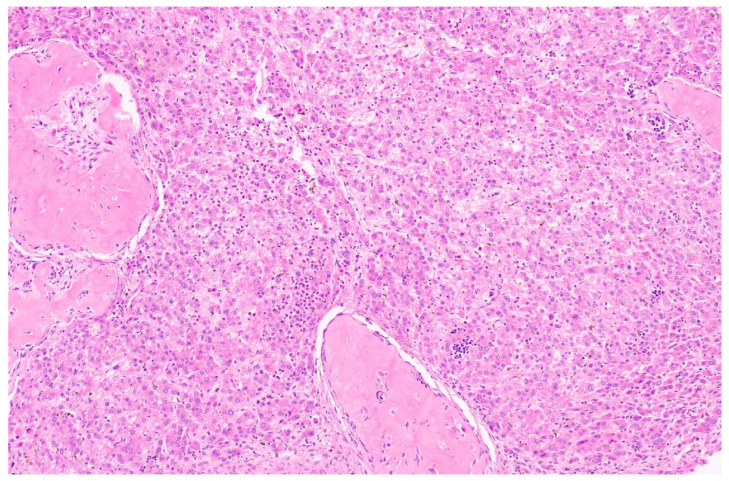
Osteoid in a hepatoblastoma admixed with the fetal epithelial cell type (H&E, ×5).

**Figure 43 diagnostics-13-03524-f043:**
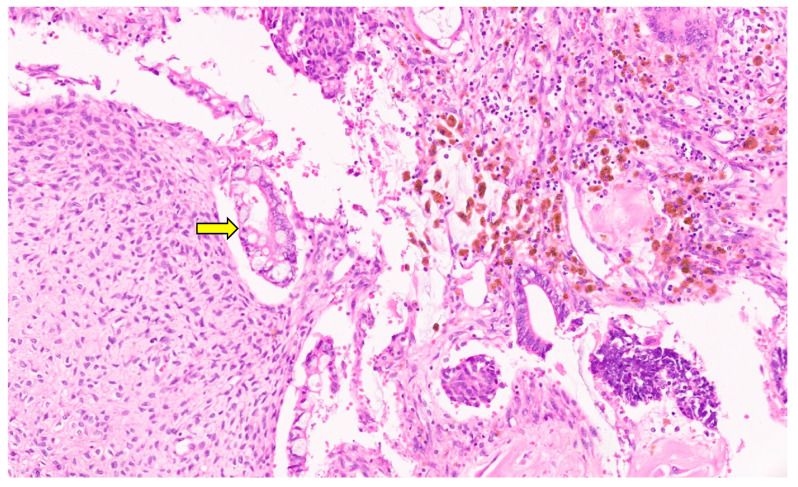
Teratoid hepatobalstoma with mucinous epithelium (arrow, H&E, ×13).

**Figure 44 diagnostics-13-03524-f044:**
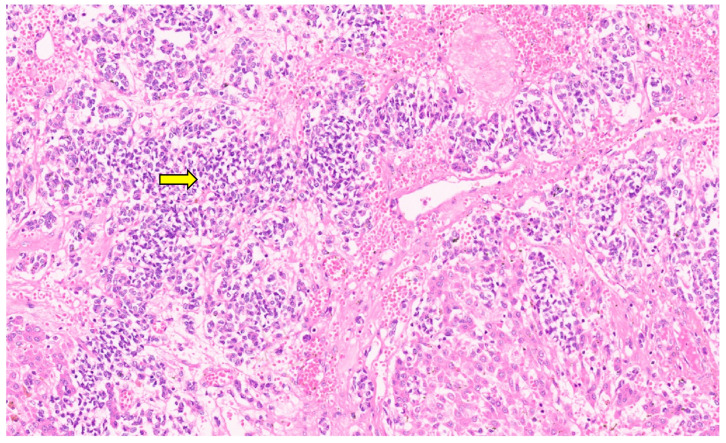
Teratoid hepatobalstoma with neuroendocrine differentiation (arrow, H&E, ×12).

**Figure 45 diagnostics-13-03524-f045:**
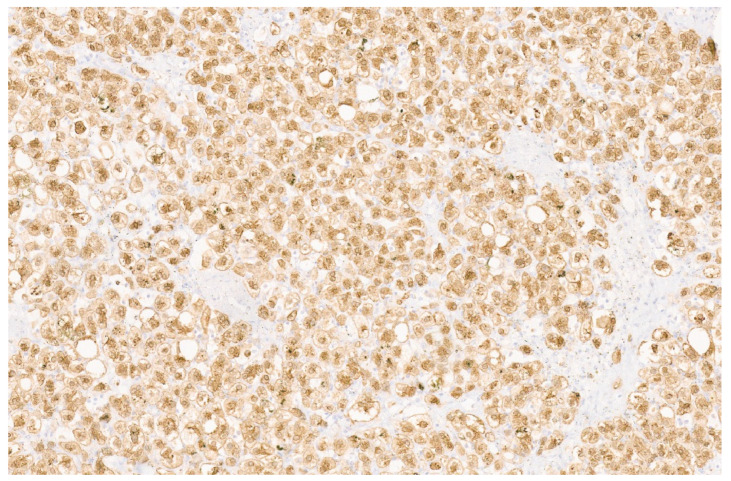
Diffuse cytoplasmic glutamine synthetase immunostaining in fetal HB (H&E, ×10).

**Figure 46 diagnostics-13-03524-f046:**
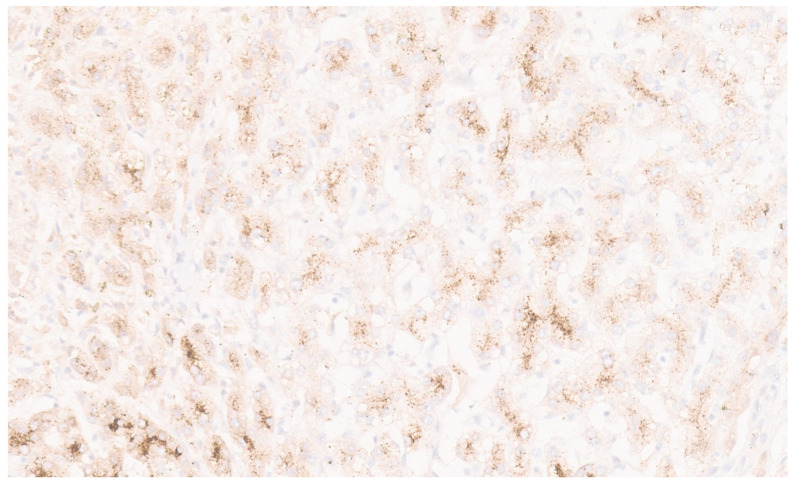
Fine ganular cytoplasmic glypican 3 immunostaining in fetal HB (H&E, ×20).

**Figure 47 diagnostics-13-03524-f047:**
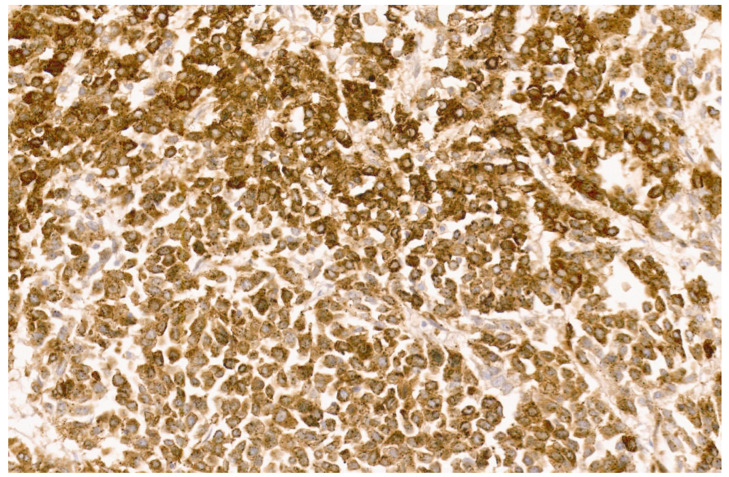
Strong coarse glypican 3 immunostaining in embryonal hepatoblastoma (H&E, ×15).

**Figure 48 diagnostics-13-03524-f048:**
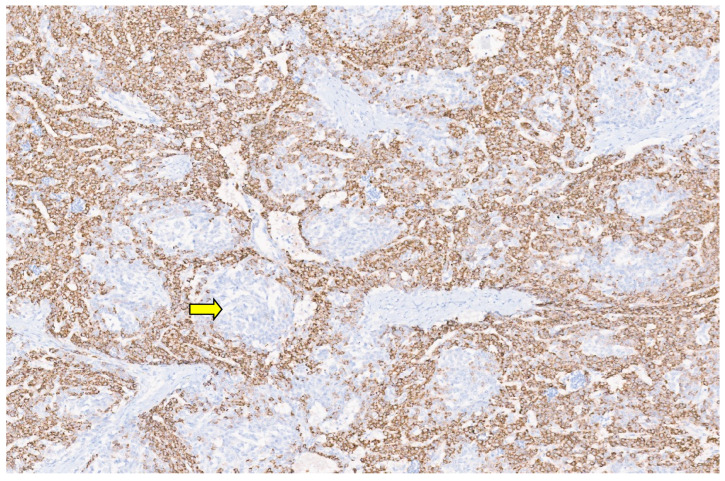
Hep-par1 loss in small-cell undifferentiated foci (arrow, ×8).

**Figure 49 diagnostics-13-03524-f049:**
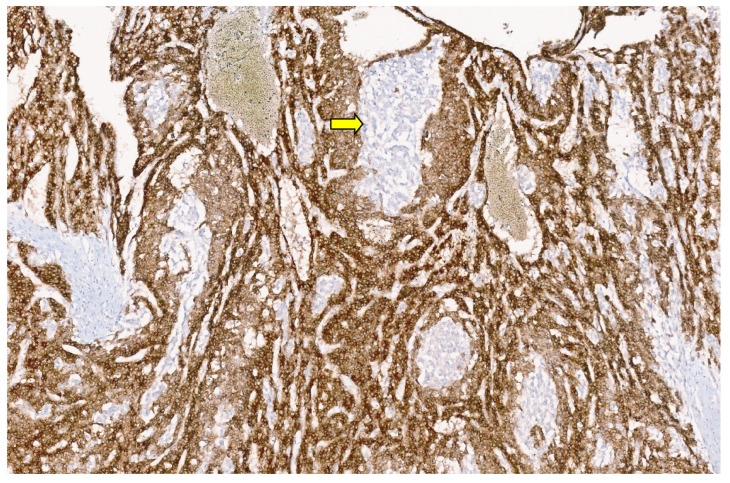
Glypican 3 loss in small-cell undifferentiated foci (arrow) with strong expression in embryonal areas (×8).

**Figure 50 diagnostics-13-03524-f050:**
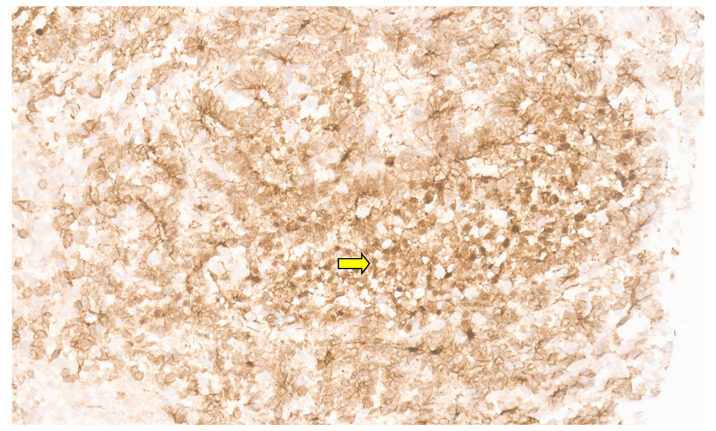
Both strong nuclear and cytoplasmic β-catenin positivity in small-cell undifferentiated foci (arrow, ×8).

**Figure 51 diagnostics-13-03524-f051:**
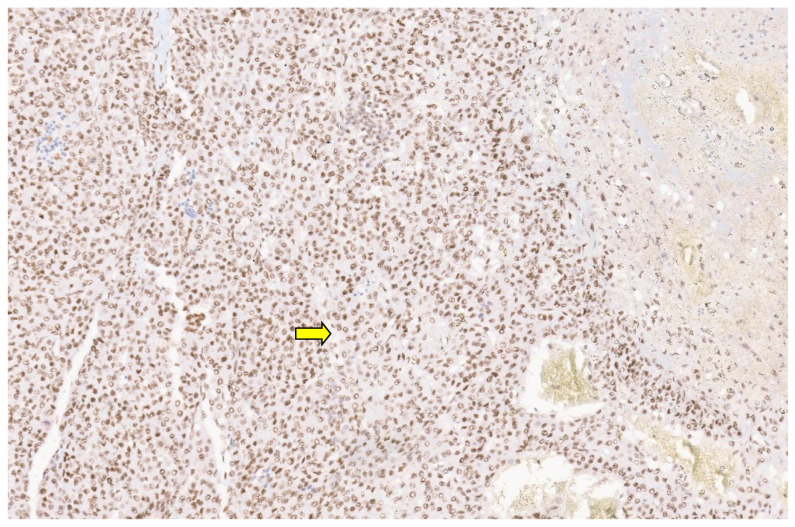
INI positivity in small-cell undifferentiated foci (arrow, ×8).

**Figure 52 diagnostics-13-03524-f052:**
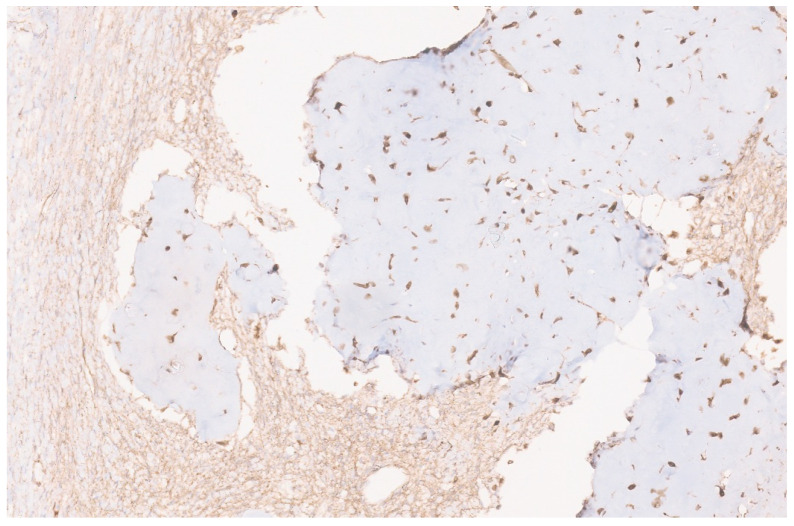
Osteoid with nuclear β-catenin immunostaining (H&E, ×10).

**Figure 53 diagnostics-13-03524-f053:**
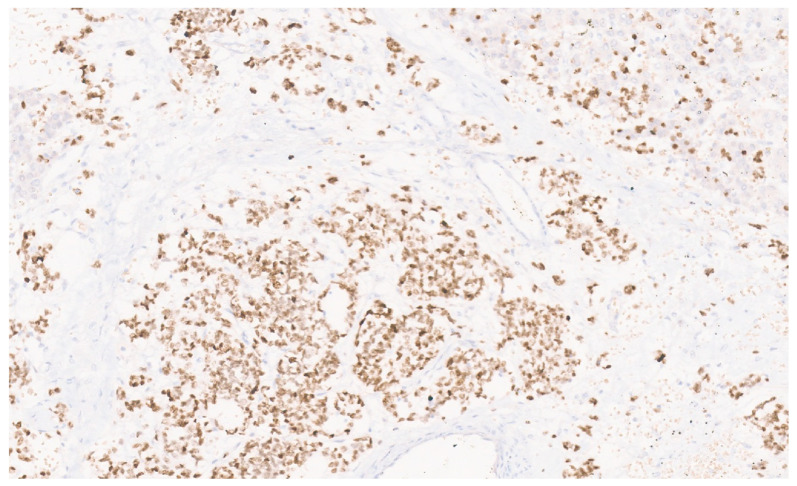
Neuroendocrine elements with positive INSM1 immunostaining (H&E, ×10).

**Figure 54 diagnostics-13-03524-f054:**
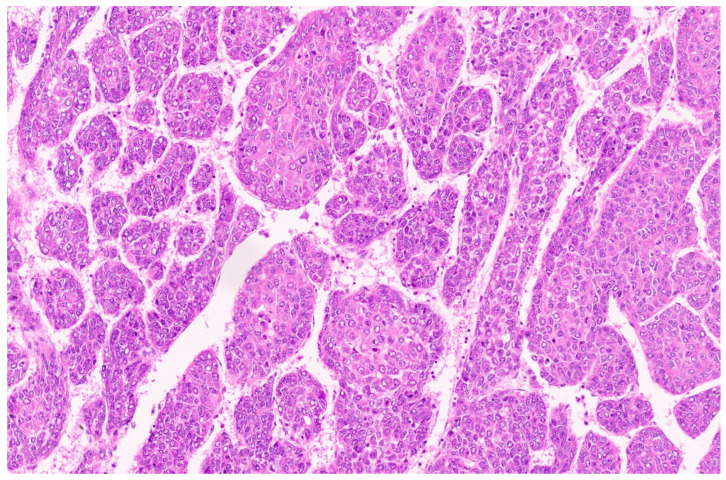
Moderately differentiated hepatocellular carcinoma with trabeculae (H&E, ×10).

**Figure 55 diagnostics-13-03524-f055:**
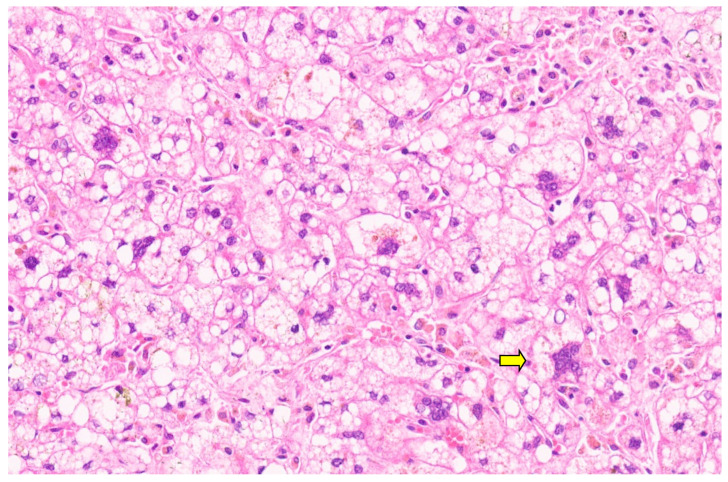
Pediatric hepatocellular carcinoma with syncytial giant cells (arrow, H&E, ×10).

**Figure 56 diagnostics-13-03524-f056:**
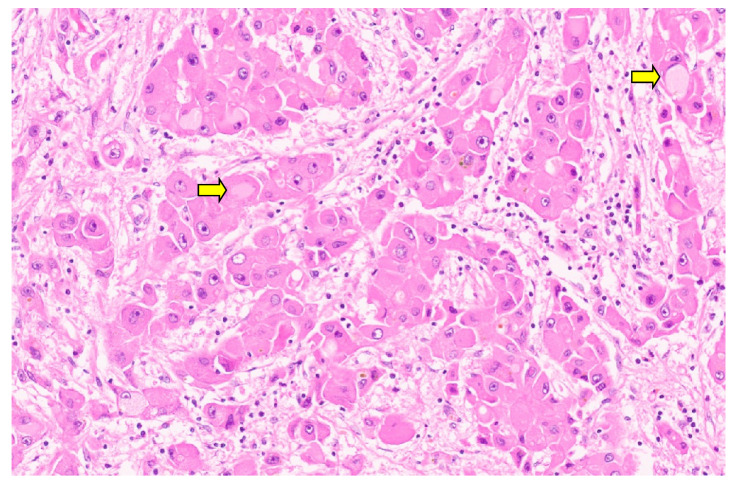
Fibrolamellar HCC with large monotonous neoplastic cells with an abundant deeply eosinophilic cytoplasm and prominent nucleoli. Pale bodies can also be noted (arrows for pale bodies H&E, ×18).

**Figure 57 diagnostics-13-03524-f057:**
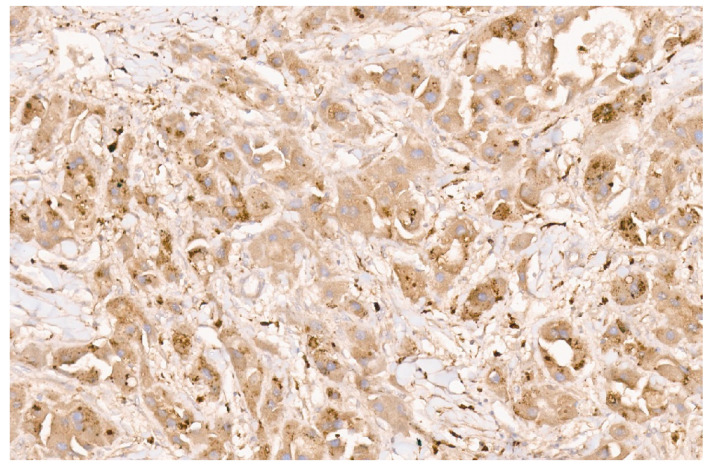
CD68 immunostaining in fibrolamellar HCC (×15).

**Figure 58 diagnostics-13-03524-f058:**
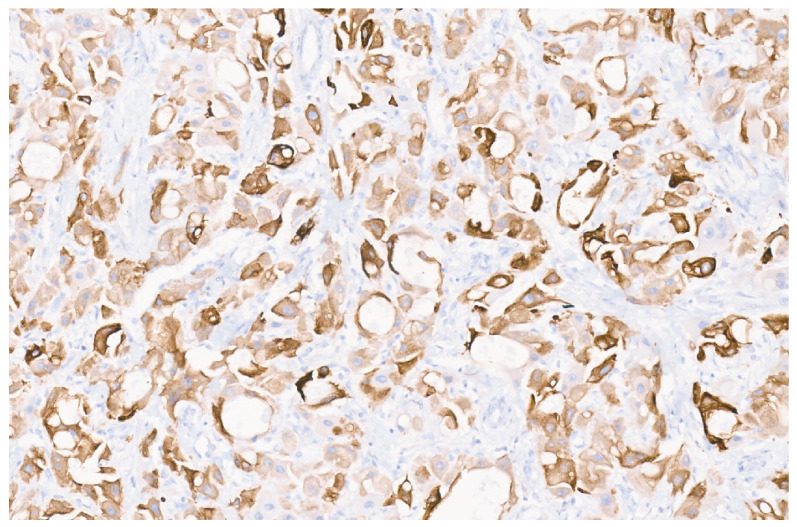
CK7 immunostaining in fibrolamellar HCC (×15).

**Table 1 diagnostics-13-03524-t001:** Pediatric liver tumors.

**Mesenchymal Tumors**
Hepatic congenital hemangioma
Hepatic infantile hemangioma
Hepatic epithelioid hemangioendothelioma
Hepatic angiosarcoma
Mesenchymal hamartoma
Inflammatory myofibrobalstic tumor
Embryonal sarcoma of the liver
Hepatobiliary Rhabdomyosarcoma
Epstein Barr Virus-associated smooth muscle tumor
**Tumors of uncertain origin**
Malignant rhabdoid tumor of the liver (INI-1 negative)
**Epithelial tumors**
Hepatic adenoma
Focal nodular hyperplasia (FNH)
Hepatoblastoma
Pediatric hepatocellular carcinoma
Fibrolamellar hepatocellular carcinoma
Hepatoblastoma with carcinoma features

**Table 2 diagnostics-13-03524-t002:** Histopathology and immunohistochemistry findings in hepatoblastoma.

Classification	Architecture, Cell Size and Shape, Mitosis	Nucleus	Cytoplasm	IHC
**Epithelial type**				
Fetal				
Low mitotic activity (well differentiated) Mitotically active (crowded fetal)	1–2-cell-thick cords of polygonal cells measuring 10–20 µ with central nuclei. Mitosis <2/10 HPF. Closely packed cells with high N/C ratio, well-delineated plasma membranes, mitosis ≥2/10HPF.	Round nucleus with finely stippled chromatin, well-delineated nuclear membrane, inconspicuous nucleoli. Round nuclei with fine chromatin.	Clear or finely granular and eosinophilic due to variable amounts of glycogen and lipids. Amphophilic cytoplasm due to decreased glycogen	Glypican 3 with finely granular cytoplasmic positivity, strong glutamine synthetase positivity, Hep par-1-positive, b-catenin membranous, cytoplasmic, with rare nuclear staining Strong coarse glypican 3 positivity, glutamine synthetase variable positivity, Hep par-1 positivity, uniform nuclear b-catenin positivity
Embryonal	Solid nests, glands, acini, pseudorosettes, or papillae. High mitotic activity level, indistinct cell borders	Enlarged angulated to oval nuclei with coarse chromatin and prominent nucleoli	Scanty dark granular cytoplasm, devoid of lipids and glycogen.	Strong coarse glypican 3 positivity, week glutamine synthetase variable b-catenin nuclear positivity, Hep par-1 positivity
Pleomorphic	Bizarre pleomorphic cells. May resemble fetal or embryonal cell types	Anisomorphic irregular large nuclei with coarse chromatin and conspicuous to prominent nucleoli	Eosinophilic to amphophilic cytoplasm	Strong coarse glypican 3 positivity, glutamine synthetase and Hep par-1 variable, strong nuclear b-catenin positivity
Small-cell undifferentiated				
SMARCB1(INI)-positive. SMARCB1(INI)-negative	Diffuse pattern, nests, round to oval cells measuring 7–8 µ. Variable mitotis with apoptosis and necrosis. Discohesive cells, resembles rhabdoid tumor	Fine nuclear chromatin with inconspicuous nucleoli. Eccentric irregular nuclei, prominent nucleoli.	Scanty cytoplasm. Abundant cytoplasmic filaments	Co expression of pan-cytokeratin and vimentin. Glypican 3, glutamine synthetase and Hep par-1 are negative
Cholangioblastic	Ductular differentiation	Large nuclei with coarse chromatin and prominent nucleoli.	Scanty dark granular cytoplasm	Nuclear b-catenin, CK7, and CK 19 are positive; glypican 3, glutamine synthetase, and Hep par-1 are negative
Macrotrabecular	The trabeculae are >5 cells thick. Variable and may resemble fetal or embryonal cell types			Strong nuclear b-cat expression
**Mixed epithelial and Mesenchymal type**				
Without teratoid features. With teratoid features.	Mesenchymal components comprise mature and immature fibrous tissue, osteoid and hyaline cartilage. Neuroectodermal-tissue-like glial tissue, neuronal cells, primitive neuroepithelium forming tubules and rosettes, retinal pigment, and squamous epithelium. Mesenchymal-tissue-like striated muscle cartilage and bone.			

**Table 3 diagnostics-13-03524-t003:** Morphological features of conventional HCC.

Histopathology	Well Differentiated	Moderately Differentiated	Poorly Differentiated
**Arrangement**	Trabeculae that are at least 3 cells thick and lined by sinusoids, pseudoacinar arrangement	Trabeculae that are 15–20 cells thick lined by sinusoids, pseudoacinar arrangement	Sheets or nests of cells not resembling hepatocytes.
**Cells and nuclei**	Resemble hepatocytes with mild nuclear pleomorphism, centrally placed round nuclei, and abundant cytoplasm	Moderate nuclear pleomorphism, large nuclei with prominent nucleoli and intranuclear inclusions, abundant cytoplasm.	High N/C ratio, nuclear pleomorphism, prominent nucleoli and intranuclear inclusions, numerous mitosis
**Necrosis**	Usually absent	May be seen	Present
**Immunohistochemistry**	Variable glypican 3 and glutamine synthetase levels. Negative for β-catenin nuclear staining	Glypican 3 and glutamine synthetase are positive in 50% of cases, β-catenin is variable	Glypican 3 and glutamine synthetase

**Table 4 diagnostics-13-03524-t004:** Summary of pediatric liver tumors.

Tumor	Age at Diagnosis	Gender Preponderance	Risk Factors	Prognosis
Hepatic congenital hemangioma	Inutero	Female predominance	-	Good
Hepatic infantile hemangioma	<12 months	Female predominance	Multigestational pregnancy, low birth weight, prematurity	Good
Epithelioid hemangioendothelioma	12 years	Female predominance	-	Variable/Uncertain
Hepatic angiosarcoma	2–7 years	Slight female predominance	Exposure to chemical carcinogens, radiation	Poor
Mesenchymal mamartoma	<2 years	Slight male predominance	-	Excellent
Inflammatory Myofibroblastic tumor	<15 years	No gender predominance	-	Good
Malignant rhabdoid tumor	<2 years	Slight male predominance	-	Poor
Embryonal sarcoma	6–10 years	No gender predominance	-	Poor
Hepatobiliary rhabdomyosarcoma	3–4 years	Male predominance	-	Poor
Epstein–Barr-virus-associated smooth muscle tumor	4 years	Female predominance	Immunodeficiency	Poor
Hepatocellular adenoma	14 years	No gender predominance	Genetic disorders, hepatic parenchymal diseases, obesity	Good
Focal nodular hyperplasia	8–11 years	Female predominance	Portosystemic shunts, chemotherapy, radiation therapy	Good
Hepatoblastoma	6 months–3 years	Slight Male predominance	Premature delivery, low birth weight	Good
Hepatocellular carcinoma	10–14 years	Slight male predominance	Hepatitis B, inherited liver diseases, biliary atresia	Poor

## Data Availability

Not applicable.

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
