# Peer review of "Update on the Pathology of Pediatric Liver Tumors: A Pictorial Review"

_diagnostics, 2023, doi:10.3390/diagnostics13233524_

Round 1

Reviewer 1 Report

Comments and Suggestions for Authors

Name of Journal: Diagnostics (ISSN 2075-4418)

Manuscript Type: Review

Title: Update on Pathology of Pediatric Liver tumors: a Pictorial Review

Manuscript ID:diagnostics-2711451

Comments

This is an interesting review to summarize the research progress of liver tumors in children. Some issues undermine the quality of the paper and should be properly addressed.

 1. Structure: Mesenchymal Tumors in Children,Vascular Tumors in Children......Mesenchymal Tumors in Children,Mesenchymal Hamartoma....... In the main text, the author classified peiatric liver tumors based on tumor histopathology and discussed their clinical presentation, imaging findings, pathology and molecular features. However, there is no histopathological classification description in the Abstract and Introduction section, which makes readers feel confused. In addition, it is recommended that the author provide a table to describe the classification of peiatric liver tumors.

 2. Terms: Inconsistent terms in subheadings, such as Pathology, Histopathology. Does “Hepatic infantile haemangioma” means Infantile hepatic hemangioma”? Most published papers describe it as Infantile hepatic hemangioma”. Please check.

 3. Introduction section: Does the introduction provide sufficient background and include all relevant references. The importance of this review is not sufficient, or the description of purpose of this article is not sufficient. Please revised.

 4. Abstract section:We discuss the clinical presentation, imaging findings, pathology, and molecular features that can help in correct identification of these tumors, which is important in managing these children. Molecular features is mentioned in the abstract, but the related descriptions of molecular features were not seen in the main text.

5. Format of Table:Please check the Format of Table 1 and the bullets and numbers in Table 1.

Author Response

Dear Reviewers and Editors,

Many thanks for your valuable comments regarding our article entitled: Update on Pathology of Pediatric Liver tumors: a Pictorial Review. We appreciate your interest and precious time spent in going through our article. We have now revised our manuscript considering the comments, critiques and questions highlighted in your reviews. We believe that the revised manuscript now reads well and fulfills the requirements for publication in Diagnostics. If you have any further queries or comments, please do not hesitate to contact us.

Best Regards

Thank you.

Mukul vij

Reviewer no 1:

This is an interesting review to summarize the research progress of liver tumors in children. Some issues undermine the quality of the paper and should be properly addressed.

  1. Structure: “Mesenchymal Tumors in Children, Vascular Tumors in Children......Mesenchymal Tumors in Children, Mesenchymal Hamartoma......”. In the main text, the author classified paediatric liver tumors based on tumor histopathology and discussed their clinical presentation, imaging findings, pathology and molecular features. However, there is no histopathological classification description in the Abstract and Introduction section, which makes readers feel confused. In addition, it is recommended that the author provide a table to describe the classification of paediatric liver tumors.

Reply: The corrections are done in the manuscript. We have revised the abstract. Table with classification of paediatric liver tumour is added in the introduction section.

  1. Terms: Inconsistent terms in subheadings, such as Pathology, Histopathology. Does “Hepatic infantile haemangioma” means “Infantile hepatic hemangioma”? Most published papers describe it as “Infantile hepatic hemangioma”. Please check.

Reply: Corrections of subheadings is done in the manuscript. Hepatic infantile haemangioma” is recommended now in the new WHO blue book on classification of paediatric liver tumours.

  1. Introduction section: Does the introduction provide sufficient background and include all relevant references. The importance of this review is not sufficient, or the description of purpose of this article is not sufficient. Please revised.

Reply: We have revised the introduction. Table with classification of paediatric liver tumour is added in the introduction section.

  1. Abstract section:“We discuss the clinical presentation, imaging findings, pathology, and molecular features that can help in correct identification of these tumors, which is important in managing these children”. Molecular features is mentioned in the abstract, but the related descriptions of molecular features were not seen in the main text.

Reply: Relevant molecular features are added in the manuscript

  1. Format of Table: Please check the Format of Table 1 and the bullets and numbers in Table 1.

Reply: Table 1 is now table 2 and is checked for bullets and numbers. corrections are done

Reviewer 2 Report

Comments and Suggestions for Authors

This is a fine and comprehensive review of pathology of liver tumors in pediatrics

The collection of histology figures is impressive and of high quality

My comments:

Figure 1: the abbreviations used need to be explained in the footnote

I need a ‘summary table’ for easy overview including for each type of tumor:

Age top

Gender preponderance

Risk factors

Frequency

Prognosis

Author Response

Reviewer no 2:

Figure 1: the abbreviations used need to be explained in the footnote

Reply: The corrections are done in the manuscript

I need a ‘summary table’ for easy overview including for each type of tumor:

Age top Gender preponderance Risk factorsFrequency Prognosis

Reply: We have added a summary table at the end of the article.

Table 4 summarizes age, gender preponderance, risk factors, and prognosis of pediatric liver tumours.

Tumour

Age at diagnosis

Gender preponderance

Risk factors

Prognosis

Hepatic Congenital Hemangioma

Inutero

Female predominance

-

Good

Hepatic infantile Hemangioma

<12 months

Female predominance

Multigestational pregnancy, low birth weight, and prematurity

Good

Epithelioid haemangioendothelioma

12 years

Female predominance

-

Variable/Uncertain

Hepatic Angiosarcoma

2-7 years

Slight female predominance

Exposure to chemical carcinogens, radiation

Poor

Mesenchymal Hamartoma

<2 years

Slight male predominance

-

Excellent

Inflammatory Myofibroblastic tumor

< 15 years

No gender predominance

-

Good

Malignant Rhabdoid Tumor

<2 years

Slight male predominance

-

Poor

Embryonal Sarcoma

6-10 years

No gender predominance

-

Poor

Hepatobiliary Rhabdomyosarcoma

3-4 years

Male predominance

-

Poor

Epstein Barr Virus-associated smooth muscle tumor

4 years

Female predominance

Immunodeficiency

Poor

Hepatocellular Adenoma

14 years

No gender predominance

Genetic disorders, hepatic parenchymal diseases, obesity

Good

Focal Nodular Hyperplasia

8-11 years

Female predominance

Portosystemic shunts, chemotherapy, and radiation therapy

Good

Hepatoblastoma

6 months – 3 years

Slight Male predominance

Premature delivery, low birth weight

Good

Hepatocellular Carcinoma

10-14 years

Slight male predominance

Hepatitis B, Inherited liver diseases, biliary atresia

Poor